# Quantifying irreversible movement in steep fractured bedrock permafrost on Matterhorn (CH)

Samuel Weber[1], Jan Beutel[2], Jérome Faillettaz[1], Andreas Hasler[3], Michael Krautblatter[4], and Andreas Vieli[1]

[1]Department of Geography, University of Zurich, Switzerland
[2]Computer Sciences, ETH Zurich, Switzerland
[3]Department of Geosciences, University of Fribourg, Switzerland
[4]Landslide Research, Technical University of Munich, Germany

*Correspondence to:* Samuel Weber (samuel.weber@geo.uzh.ch)

**Abstract.** Understanding rock slope kinematics in steep fractured bedrock permafrost is a challenging task. Recent laboratory studies have provided enhanced understanding of rock fatigue and fracturing in cold environments but were not successfully confirmed by field studies. This study presents a unique time series of fracture kinematics, rock temperatures and environmental conditions at $3500\,\mathrm{m}$ a.s.l. on the steep, strongly fractured Hörnligrat of the Matterhorn (Swiss Alps). Thanks to eight years of continuous data, the longer-term evolution of fracture kinematics in permafrost can be analyzed with an unprecedented level of detail. Evidence for common trends in spatio-temporal pattern of fracture kinematics could be found: A partly reversible seasonal movement can be observed at all locations, with variable amplitudes. In the wider context of rock slope stability assessment, we propose to separate reversible (elastic) components of fracture kinematics, caused by thermo-elastic strains, from the irreversible (plastic) component due to other processes. A regression analysis between temperature and fracture displacement shows that all instrumented fractures exhibit reversible displacements that dominate fracture kinematics in winter. Furthermore, removing this reversible component from the observed displacement enables to quantify the irreversible component. From this, a new metric – termed index of irreversibility – is proposed to quantify relative irreversibility of fracture kinematics. This new index can identify periods when fracture displacements are dominated by irreversible processes. For many sensors, irreversible enhanced fracture displacement is observed in summer and its initiation coincides with the onset of positive rock temperatures. This likely indicates thawing related processes, such as melt water percolation into fractures, as a forcing mechanism for irreversible displacements. For a few instrumented fractures, irreversible displacements were found at the onset of the freezing period, suggesting that cryogenic processes act as a driving factor through increasing ice pressure. The proposed analysis provides a tool for investigating and better understanding processes related to irreversible kinematics.

**Keywords**

Fracture kinematics, steep bedrock permafrost, high mountain permafrost, fracture monitoring

# 1   Introduction

On steep high-alpine mountain slopes, the behavior of frozen rock masses is an important control of slope stability when permafrost warms or thaws and seasonal frost occurs. During the summer heat wave 2003, air temperatures across a large portion of Europe were $3°C$ higher than the 1961–1990 average (Schär et al., 2004), causing deep thaw and coinciding with exceptional rockfall activity in the European Alps (Gruber et al., 2004). In the last century, the upper tens of meters of Alpine permafrost in Europe have been warmed by $0.5 - 0.8°$C (Harris et al., 2003). Assuming that warming will continue or even accelerate, rock slope instabilities are expected to become increasingly important for scientists, engineers and inhabitants in the vicinity of high mountain permafrost regions (Gruber and Haeberli, 2007; Keuschnig et al., 2015). A coexistent growth of vulnerable socio-economic activities in alpine areas potentially leads to rising risk (Jomelli et al., 2007). In the USA and Europe, global gravity-driven slope instabilities cause damage in the range of billions of euros each year (Sidle and Ochiai, 2006). Improved monitoring strategies and hazard assessment for the dynamics of frozen rock walls are therefore needed and require better understanding of processes and factors controlling stability of potentially hazardous slopes.

Terzaghi (1962) postulated that the stability of steep unweathered rock slopes is determined by the mechanical defects of the rock such as joints and faults and not by the strength of the rock itself. In cold regions, rock is exposed to frost cycles of variable length, leading to mechanical rock damage caused by different processes, such as thermal gradients (Hall et al., 2002) or cryostatic pressure (Walder and Hallet, 1985). Ice formation is therefore an important driver of rock fracturing and can be produced by ice expansion or ice segregation. These two processes have been widely discussed, but it remains difficult to integrate this knowledge with field observations (Matsuoka and Murton, 2008). Assessing and anticipating rock wall stability is a challenging task, mainly because of the incomplete understanding of precursory signals and the inherent mechanical complexity of fractured inhomogeneous rock and ice masses (Arosio et al., 2009). Surface displacement measurements have been applied in several studies to survey fracture kinematics in permafrost revealing a clear reversible component related to thermal expansion (Wegmann and Gudmundsson, 1999; Matsuoka and Murton, 2008; Nordvik et al., 2010; Hasler et al., 2012; Blikra and Christiansen, 2014). Often, an additional irreversible displacement component is observed, which is relevant for the stability assessment of potentially hazardous slopes, but has so far not been thoroughly quantified in existing studies. In this study and based on a new eight year continuous data set of fracture kinematics, we propose and apply a methodology for separating and quantifying such irreversible displacements.

## 1.1   Permafrost rock slope kinematics and environmental controls

Fracture displacements, reversible and irreversible, is controlled by a variety of processes and external environmental forcing which are outlined in Fig. 1 and discussed in more detail in this section. The schematic in Fig. 1a combines the concept of destabilization by warming ice-filled rock joints developed by Gruber and Haeberli (2007), the rock-ice-mechanical model by Krautblatter et al. (2013) and the permafrost controlled rock slide model by Blikra and Christiansen (2014), in which topographically controlled thermally induced stresses, ice and water pressure act as driving processes. The resisting mechanisms are shear resistance and fracture infill. The shear resistance is given by cohesive rock bridges, ice deformation/fracture that reduces

stresses through plastic work and cohesion/friction along fractures. All processes strongly depend on temporal fluctuating environmental forcing as well as the static geological or geotechnical characteristics. Many of these processes interact and result in complex combinations of individual contributions. The observed fracture kinematics usually consists of a reversible (elastic) and irreversible (plastic, creep and rupture) component. The corresponding specific relations between fracture kinematics and temperature are indicated in more detail below (see plots in Fig. 1b).

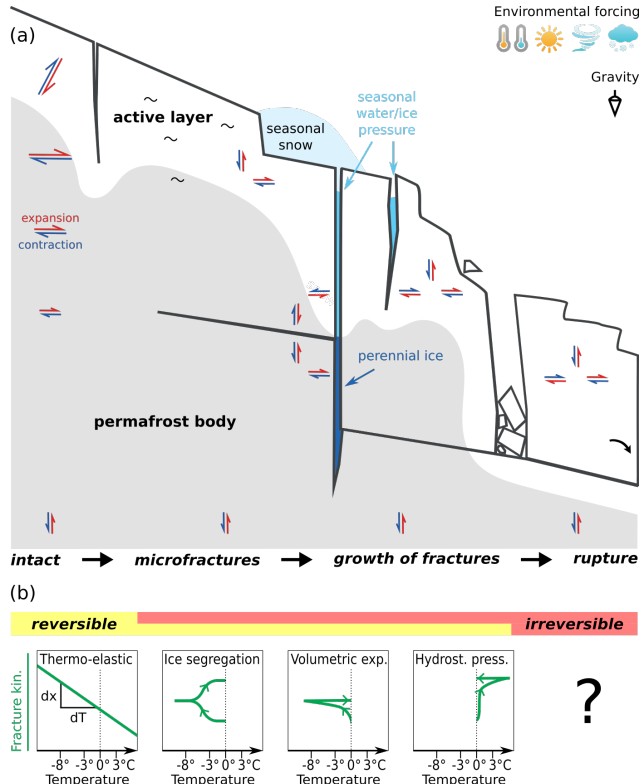

**Figure 1.** Schematic visualization of kinematics in steep fractured bedrock permafrost shows the main acting mechanisms influenced by varying environmental forcing. (a) The gray area indicates permafrost, which is thermally defined as ground with a temperature below $0°$ C for at least two consecutive years. The overlaying active layer is exposed to sub-annual freezing and thawing. (b) The indicated mechanisms can lead to fracture kinematics and each isolated mechanism causes specific movement patterns, illustrated with the schematic plots showing the relation between fracture kinematics and rock temperature.

**Thermally induced stress**

Rock tends to expand on warming and to contract on cooling and results in a reversible displacement behavior. Assuming homogeneous thermal conditions, a change in length $\Delta L$ of rock in all directions can be described by a linear function of

temperature:

$$\Delta L = L_0 \cdot \alpha \cdot \Delta T \tag{1}$$

where $L_0$ is the initial length, $\alpha$ the material dependent linear expansion coefficient and $\Delta T$ the temperature change of the material. In laboratory experiments, Wolters (1969) showed a linear strain-temperature relation for different rocks (marly limestone, limestone, claystone, granite and basalt) between $-20$ and $+80°$ C. Short-lived thermo-elastic strains accommodate volume changes as displacements, typical for fractured bedrock in non-permafrost (Watson et al., 2004) as well as in permafrost areas (Hasler et al., 2012). This is therefore a reversible mechanism as it is driven by cycling temperature. Equation 1 is a highly simplified approximation and ignores: (i) anisotropy and heterogeneity of the rock mass, (ii) complex 3D temperature regimes, (iii) the unknown behavior of fractured bulk rock masses and (iv) a potential non-linear expansion coefficient of rocks containing ice-filled pores (Jia et al., 2015). However, several studies in permafrost bedrock with different measurement setups (e.g. Wegmann and Gudmundsson, 1999; Matsuoka, 2001; Matsuoka and Murton, 2008; Nordvik et al., 2010) confirm a simple relation between fracture kinematics and (rock-) temperature at different time scales ranging from diurnal to annual. Further, Nordvik et al. (2010) applied a multiple regression analysis with aggregated sinusoidal air temperature to model the seasonal fracture kinematics and propose this approach for predictions of fracture kinematics in the context of early warning systems.

Thermally induced stress may cause rock fracture either by repetitive low-magnitude temperature cycles that lead to thermal stress fatigue or by a rapid temperature change (Murton, 2007). This might lead to irreversible displacement.

**Cryogenic kinematics during freezing periods and related kinematics during warming**

Kinematics in partly frozen rock masses may also be caused by increasing ice pressure evolving in ice-filled fractures or pores by cryogenic processes. Volumetric expansion or ice-segregation are the most common explanations here. Volumetric expansion in laboratory experiments is only effective if freezing leads to sealing of rock fractures or porous samples before ice can extrude (Davidson and Nye, 1985). However volumetric expansion also applies in pores which are on average saturated by much less than $91\%$. Due to the heterogeneous moisture distribution, some pores will always have a higher saturation and thus have insufficient space for the volumetric expansion of freezing water (Jia et al., 2015). Ice segregation, which is most effective between $-3°$ and $-6°$ C with sustained water supply (Hallet et al., 1991), describes the freezing of the migrated water at the freezing site, which results in lenses or layers of segregated ice due to ice growth (Matsuoka and Murton, 2008). Ice formation induces pressure variations in rock pores and cracks at a level that is sufficient to crack intact high porosity rocks (Murton et al., 2006). Based on numerical simulations, ice segregation can even occur in low porosity rocks in an estimated temperature range from $-4$ to $-15° C$ if liquid water is available (Walder and Hallet, 1985). In nature, conditions required for ice segregation are more commonly met than the conditions required for volumetric expansion. It has to be considered that ice pressure and its release by melting can also produce reversible fracture displacements.

While ice-filled joints can form relatively tough ice bodies at low temperatures, the shear resistance decreases with rising temperature and reaches a minimum just below the thawing point (Davies et al., 2001). Mellor (1973) showed a significant reduction in strength when intact water-saturated rock thaws. Periodic loading of discontinuities due to thermo-mechanical

effect acts as a mesoscale fatigue process. This can result in enhanced displacement and progressive rock slope failure (Gischig et al., 2011). After a certain fatigue life, tensile and compressive strength reduce to residual values (Jia et al., 2015). Besides the relatively slow process of heat conduction, the warming of frozen fractured bedrock is influenced by advective heat transport by percolating water. This process efficiently transfers heat from the surface to fractures (Hasler et al., 2011). Such advective

heat transport produces rapid variations in mechanical properties, which can potentially deform frozen discontinuities and consequently prepare rock-slope failures. But the potential formation of basal ice layers between the snow and the rock prevents percolation of snow melt water into fractures (Phillips et al., 2016).

**Hydro kinematics occurs during summer months and during snow melt**

Irreversible displacement caused by water-related processes can only be observed in summer, because the availability of liquid

water is very limited during winter. Water can increase the effective stress through hydrostatic pressure but leave the strength (i.e. cohesion and friction) untouched, whereby hydrostatic pressure is mostly determined by the height of the water column. It depends amongst other factors on the hydraulic permeability of the rock mass. Hydraulic permeability is much lower in rock masses with frozen and ice-filled fissures than unfrozen fissures and often causes high hydrostatic stress due to perched water (Pogrebiskiy and Chernyshev, 1977). But there are no detailed empirical quantitative studies on how hydrostatic pressure

affects rock walls in permafrost regions (Krautblatter et al., 2013). However, hydrostatic pressure is presumed not to dominate in the near-surface layer of strongly fractured steep bedrock, where the ability for drainage is quite high. However, changing conditions in shear zones, e.g. from dry to wet, can lead to irreversible displacement, for example caused by water (melting snow or rain) percolating through preexisting fissures. Even with low hydrostatic pressure, the presence of water can reduce cohesion in fine-grained material containing clay and is expected to have a strong influence in fractures filled with fine-grained

material.

**Long term evolution**

In the long term, displacements along fractures act to change the persistent gravitationally-induced stress distribution in the rock mass controlled by the bulk material stiffness and rock mass strength properties. Deformation and fracture of ice can absorb stress along fractures and lead to dislocation (Matsuoka, 1990) while fracture infill by debris or fine grained material

can significantly alter shear resistances of fractures in a frozen or unfrozen state. Persistent reversible thermo-elastic oscillations of an initially stable rock mass (stable phase in Fig. 2), in combination with an increase in shear stress due to concentration of stress at rock bridges or a decrease in shear resistance, leads to irreversible surface displacement (unstable phase in Fig. 2). Therefore, irreversible displacements could be a first indication for the initiation of rock slope failure.

    However, reversible and irreversible displacements are often superimposed and it is difficult to interpret kinematics data and

relate them to external forcing. Furthermore, failure of heterogeneous natural materials often results from the culmination of progressive irreversible damage involving complex interactions between multiple defects and growing microcracks (Faillet-taz and Or, 2015). Therefore quantifying the irreversible component of the overall fracture displacement is expected to give valuable information in the context of rock slope stability assessment (Fig. 1).

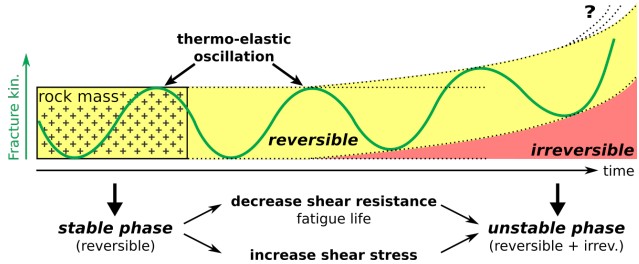

**Figure 2.** Evolution of a permafrost affected rock mass with persistent thermo-elastic oscillations: initially reversible displacement of rock mass can develop an additional irreversible component either by an increase in shear stress or by a decrease in shear resistance.

## 1.2 Aim of this study

This study focuses on the kinematics of fractured bedrock permafrost (Fig. 1a). It aims at quantifying irreversible fracture displacements in relation to environmental forcing. For this, the reversible (elastic) components of fracture displacement, due to thermo-elastic strains, are separated from the irreversible (plastic) component, due to other processes. Using a statistical model for the reversible component, we are able to investigate the kinematics in fractured bedrock permafrost with a focus on enhanced opening and shearing of fractures. The term displacement used in the following refers to the movement of one side of the fracture with respect to the other. Irreversible displacement refers to slow rock slope deformation, which could be seen as a part of slope instability, potentially preparing slope failure. The statistical model introduced here has been developed and tested on the base of eight years of continuous high resolution temperature and fracture kinematics measurements from the Matterhorn Hörnligrat, a high mountain permafrost monitoring site. This study addresses three main questions:

1. How can we statistically separate reversible from irreversible fracture kinematics?

2. Is there a common inter-annual pattern of irreversible fracture displacements in all instrumented fractures?

3. Under what environmental conditions do enhanced irreversible fracture displacements occur?

## 2 Site description, instrumentation and field data

The relative fracture displacement and thermal conditions were measured at Matterhorn Hörnligrat (Swiss Alps) at an elevation of $3500\,\mathrm{m}$ a.s.l. (see Fig. 3) using the experimental setup by Hasler et al. (2012). The field site is suitable for such measurements due to: (1) the occurrence of ice-filled fractures indicated by an ice-containing scarp after a block fall event (approx. $1500\,\mathrm{m}^3$) in summer 2003, (2) strong fracturing and (3) a large gradient of surface thermal conditions allowing installation of thermistors and crackmeters at locations with contrasting conditions (cf. Hasler et al., 2012).

This field site consists of spatially heterogeneous steep fractured bedrock with partially debris covered ledges. The mean annual air temperature is $-3.7^\circ\mathrm{C}$ for the time period $2011-2012$ (see Fig. 11 in appendix A). The precipitation mostly falls as snow with occasional infrequent rainfall events in summer. Winter temperatures (down to $-27^\circ\mathrm{C}$ in $2011-2012$) in

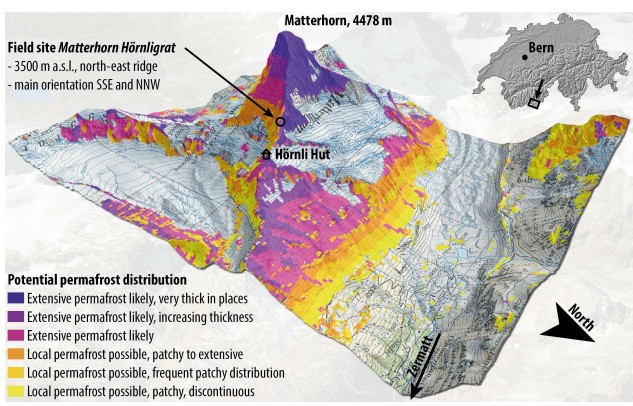

**Figure 3.** 3D overview of the Hörnligrat field site on the north-east ridge of the Matterhorn, in Valais, Switzerland (based on map.geo.admin.ch, Google Earth and SRTM). Colors indicate the potential permafrost distribution (FOEN, 2005). At this field site, extensive permafrost with a thin active layer is expected on the north side of the ridge. On the south side of the ridge, local permafrost is possible with a considerable active layer.

combination with exposure to strong wind (up to $88\,km/h$ in $2011-2012$) results in a preferential snow deposition in fractures, on ledges and at other concave micro-topographical features, which can be observed using the webcam images (see Fig. 4). On the south side the accumulated firn disappears completely during summer, while on the north side snow patches persist all year round. These factors lead to a complex temperature regime and therefore need a correspondingly large amount of precisely
5 measured data (Krautblatter et al., 2012).

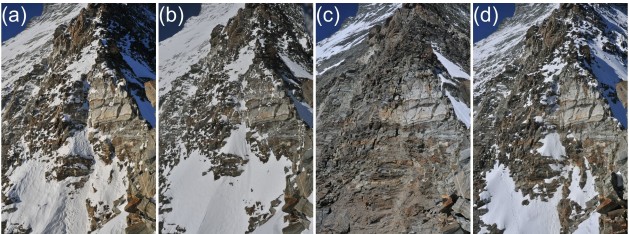

**Figure 4.** Four webcam pictures, taken in the morning on (a) 01 Jan 2015, (b) 03 Apr 2015, (c) 01 Jul 2015 and (d) 01 Oct 2015, illustrate the varying snow deposition patterns.

In this study three types of data were recorded at different locations: relative fracture displacements perpendicular to and along fractures at $2\,min$ intervals (temperature compensated, accuracy of $\pm0.01\,mm$ over entire temperature range), temperature at different depths in rock and in fractures at $2\,min$ intervals (accuracy of $\pm0.2°\,C$) and meteorological data using a Vaisala WXT520 weather station (location *mh25* in Fig. 5). The time series of the weather station is interrupted for brief periods
10 (several weeks) due to technical problems with the electronics, but a complete continuous time series is available for the years

2011 and 2012. Seven high resolution images per day (12.0 MP, giving an approximate pixel resolution of 1.5 cm) serve for visual inspection of the instrumentation and also provide information on snow deposition.

Figure 5 gives a spatial overview of all installations and measurement locations. Basic meta information of the measurement locations is given in Table 1 for all locations. Displacements perpendicular to the fracture are measured at locations *mh02–*

*mh04* while displacements perpendicular and parallel to the fracture are measured at locations *mh06*, *mh08* and *mh20–mh22*. Crackmeter at location *mh01* is installed next to a fracture on a rock mass with several microcracks (sub-millimeter scale). Temperature in fractures at different depths are available at all crackmeter locations, except at locations *mh20–mh22*. Rock temperature at different depths $(0.1 - 0.85\,\mathrm{m})$ is measured at the additional locations *mh10–mh12*. All sensors are embedded in a low power wireless sensor network that provides all year-round data at near real-time (Beutel et al., 2009). The observed

temperature and fracture kinematics measurements were aggregated as 10 min averages to reduce noise. A detailed description and explanation of the measurement setup is given by Hasler et al. (2012, Section 3).

Instrumentation started in autumn 2007 and continuous time series are available since summer 2008 for locations *mh02*, *mh03* and *mh06*. The measurement network was extended in Summer 2010 with additional sensors and by establishing new measurement locations (*mh01*, *mh04*, *mh08* and *mh20–mh22*). This results in up to eight years of data for rock and fracture

temperatures, fracture kinematics and environmental conditions.

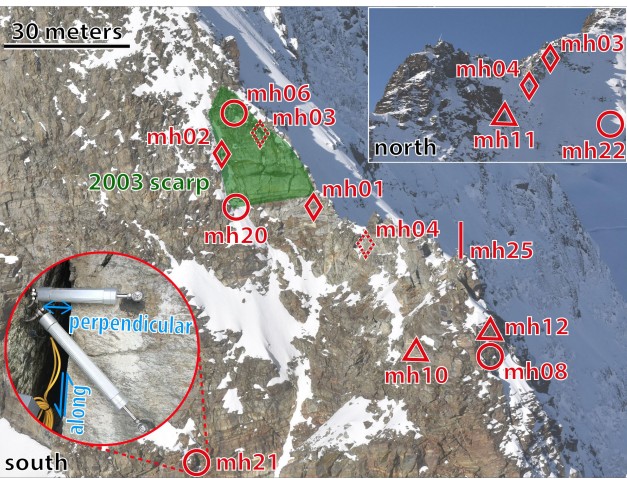

**Figure 5.** Overview of crackmeter installations. Location *mh01–mh04* (indicated with ◇) are instrumented with one crackmeter perpendicular to the fracture. Location *mh06*, *mh08* and *mh20–mh22* (indicated with ◯) are instrumented with two crackmeters to calculate displacements perpendicular to and along fracture. Temperature measurements in fractures exist at most location. Locations with only rock temperature measurements are indicated with △ while for the weather station | is used. Scarp of the 2003 rockfall is shaded green.

**Table 1.** Meta information for all measurement locations providing characteristics, type, orientation and instrumentation. If type is "fracture", thermistors are installed in fracture. Otherwise the thermistors are drilled in rock.

| Location | Characteristics | Type | Aspect | Slope | Crackmeter | Depth of thermistors $T1$, $T2$, ... (m) |
|---|---|---|---|---|---|---|
| mh01* | intense solar radiation, microcracks | fracture | 95° N | 75° | 1 axis | 0.1, 0.4, 0.7, 0.5 |
| mh02† | concave, often snow, wet | fracture | 80° N | 50° | 1 axis | 0.1, 0.3, 0.4 − 0.8 [3, 1, 2] |
| mh03 | lower part snow | fracture | 350° N | 65° | 1 axis | 0.1, 0.4, 0.6 − 0.8 [5] |
| mh04 | saddle north | fracture | 320° N | 70° | 1 axis | 0.05, 0.2, 0.2 − 0.5 [3, 1] |
| mh06 | corner, often snow | fracture | 90° N | 60° | 2 axes | 0.1, 0.8, 1.5, 1.8 |
| mh08 | wide, ventilated, close to ridge | fracture | 50° N | 90° | 2 axes | 0.1, 1, 2, 3 |
| mh10 | intense radiation, fracture 1 m beside | rock | 140° N | 90° | — | 0.1, 0.35, 0.6, 0.85 |
| mh11 | occasionally snow, no fracture | rock | 340° N | 70° | — | 0.1, 0.35, 0.6, 0.85 |
| mh12 | snow free, fracture beside | rock | 45° N | 85° | — | 0.1, 0.35, 0.6, 0.85 |
| mh20 | corner, often snow, wet | fracture | 70° N | 70° | 2 axes | — |
| mh21 | wide, south side | fracture | 70° N | 85° | 2 axes | — |
| mh22 | wide, north side | fracture | 70° N | 85° | 2 axes | — |

\* installed next to a fracture across microcracks

† rock instrumented broke off completely during a bad weather period (14 August 2015)

[X] number in square brackets indicates number of thermistors in the given depth range without exact depth information

$X$, [X] depth information or number in gray indicates problems with thermistor

## 3    Data analysis method

### 3.1    Correlation analysis

In a first step, we investigate the linear relation between fracture displacements and temperature. We looked for a time period, during which fracture kinematics are best described by temperature. For the evaluation of these temperature dependent fracture kinematics, we compute the Pearson correlation (LeBlanc, 2004, p. 292) for varying time periods (different start time and duration). Each location instrumented with crackmeters is individually correlated with all available fracture and rock temperature data (depths of used thermistors are indicated black in Table 1). As additional constrain time periods (1) have to be at least 70 days, (2) have to be in the time window between 1 Oct 2013 and 1 Jan 2015 (complete data availability at all instrumented locations) and (3) the temperature range must exceed 8° C. This optimal time period is determined independently for displacements perpendicular and along fractures.

### 3.2    Linear regression model (LRM)

In a second step, we aim to reproduce the reversible component of fracture kinematics caused by thermo-elastic strain. For each measurement location, the linear regression function and its parameters are computed for the optimal time period (trainings phase) determined by the correlation analysis (see Section 3.1). The linear regression model (LRM) applies this function with

temperature $T$ [$^\circ$C] for the complete time series to reproduce the reversible fracture displacement $y_{\text{rev}}$ [mm]:

$$y_{\text{rev}} = \beta_0 + \beta_1 \cdot T + e \tag{2}$$

where intercept $\beta_0$ [mm] and slope $\beta_1$ [mm/$^\circ$C] are the regression parameters and $e$ [mm] is the residual. This model is based on the assumption of a constant linear elastic rheology in the considered temperature range for all consecutive years. Irreversible kinematics is assumed to be negligible during the trainings phase. Note that the LRM is applied indistinctly perpendicular or along fracture.

## 3.3 Irreversibility index

We build a metric (termed irreversibility index) that aims at detecting periods during which overall kinematics is not dominated by thermo-elastic strains. This index uses the absolute difference ($\Delta y$) between the observed fracture data ($y_{\text{obs}}$) and the modeled reversible fracture kinematics component ($y_{\text{rev}}$) given by the LRM as input:

$$\Delta y = |y_{\text{obs}} - y_{\text{rev}}| \tag{3}$$

Finally, index $I$ is calculated applying the following function to $\Delta y$:

$$I = (\mu + 2 \cdot \sigma) - (\mu - 2 \cdot \sigma) = 4 \cdot \sigma \tag{4}$$

where the sliding functions $\mu$ (mean) and $\sigma$ (standard deviation) are evaluated over all data points in the past $21$ days. The length of the sliding window is a trade off between high noise-level and loosing important signals due to smoothing. The two standard deviation range considers $95\%$ of data around mean and thus ignores outliers. The output value of the irreversibility index is a positive number of unit mm/year. A value of zero means that the displacement is fully reversible. The higher the number, the higher the proportion of irreversibility.

## 3.4 Thawing degree days (TDD) and fracture kinematics summer shift (SHT)

In order to put the fracture kinematics data in context of thawing or freezing, we use the concept of thawing degree days (TDD). The TDD concept takes into account the amount of energy available for thawing/melting over the course of the year (Huybrechts and Oerlemans, 1990). It is here used as a rough approximation of the total energy available for melting ice or thawing permafrost. The thawing degree day sum (TDD) is defined as the total sum of daily average rock temperature above $0^\circ$ C over one year.

The fracture kinematics summer shift $y_{\text{SHT}}$ represents the shift in kinematics between two consecutive winters and is calculated as:

$$y_{\text{SHT}} = \overline{y}_{\text{obs,winter}^+} - \overline{y}_{\text{obs,winter}^-} \tag{5}$$

with the mean fracture kinematics during winter given by

$$\overline{y}_{\text{obs,winter}} = \sum_{k=t_1}^{t_2} y_{\text{obs}}/n \tag{6}$$

where $t_1 = 1\,Nov$ and $n$ the number of measurements. The end time $t_2$ is usually defined by a fix date $t_2 = 1\,May$ unless the rock temperature rises above a defined threshold value of $-1^\circ$C before this date. If this is the case, the end time is given by the date when the rock temperature reaches this threshold $(t_2 = \mathrm{date}(T_\mathrm{rock} < -1^\circ\mathrm{C}))$.

## 4    Results and interpretation

Figure 6 shows the rock temperatures at $85\,\mathrm{cm}$ depth for different aspects (a) and the fracture displacements, relative to the start of the measurements, for all locations perpendicular to the fractures (b) and along the fractures (c). Partly reversible fracture displacement can be observed at all locations with different seasonal movement amplitudes, except for location *mh02*. Most of them also show a long term trend indicating an additional irreversible component of variable magnitude and sign. The individual displacement pattern of each location may be influenced by differences in geometric mesoscale arrangement of rock, where different combinations of processes dominate. An irreversible displacement is indicated at most locations in early summer (e.g. *mh02–mh04*, *mh06*, *mh08* and *mh20*) but the exact timing and pattern is difficult to quantify. The fracture displacements of *mh02* and *mh20* are not visible after mid 2015 as they are out of range (Fig. 6). This abrupt and large displacement is due to a small rock fall event with a volume of a few cubic meters on 18 May 2015. The functionality of both crackmeters was however not affected. But the thermistors at location *mh02* were damaged by falling rocks. Hence the temperature time series ends on 18 May 2015. After this rock fall event, the fracture at location *mh02* continued to deform in several small steps until late summer (14 August 2015) when the instrumented rock broke off completely during a bad weather period (see Fig. 12). The observed variable spatial and temporal patterns in fracture displacements (Fig. 6) indicate that a field site cannot be described by a single measurement location and a short measurement period. Therefore, longterm monitoring of several fractures is essential to observe different modes of kinematics and accordingly to improve the process understanding of the fracture kinematics.

In the following paragraph, we present the analysis of a set of 3 locations in more detail, namely *mh02* (South), *mh03* (North) and *mh08* (East, on ridge). These locations were selected according to their contrasting modes of kinematics and their variations in aspect and cover all different patterns of observed fracture displacements.

### 4.1    Regression analysis of fracture displacement with temperature

The time periods during which fracture displacements exhibit best correlation with temperature are shown in Table 2 and have a typical duration of three to 5 months. The variation in length of 1–2 weeks results in similar correlation coefficients. The regression analysis between temperature and fracture kinematics (perpendicular to and along fracture) shows negative correlation coefficient between $-0.90$ and $-0.99$ for all instrumented fractures. The fracture displacements at most locations correlate best with rock temperatures at $0.85\,\mathrm{m}$, while the correlation with the other available rock temperatures are much lower. Only a few instrumented fractures correlate best with fracture temperatures (between $0.2$ and $0.8\,\mathrm{m}$). In general, all determined time periods for fracture kinematics perpendicular to fracture are in winter or early spring. The time periods for

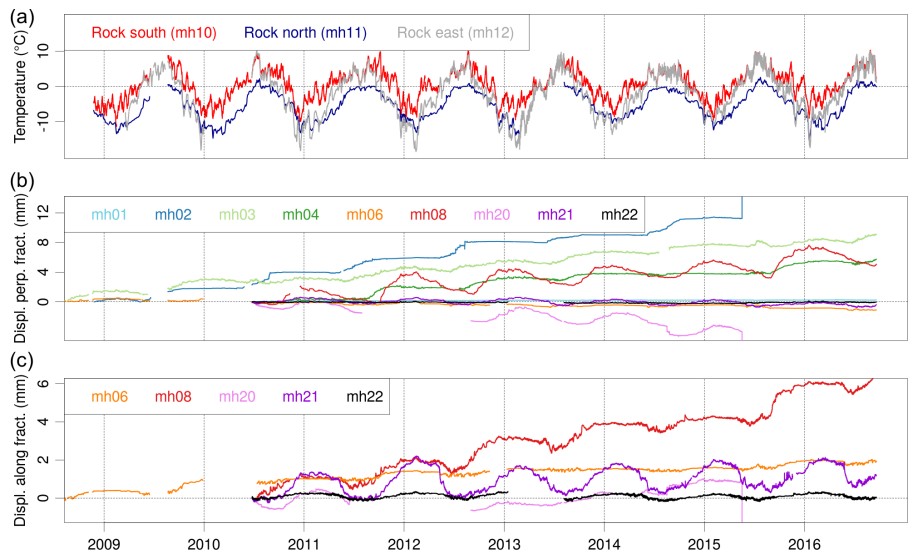

**Figure 6.** Thermal conditions and fracture displacements at the Matterhorn Hörnligrat field site over a course of eight years: (a) The thermal conditions are shown by characteristic rock temperatures for the south, east and north side of the ridge measured at a depth of $0.85\,\mathrm{m}$. The fracture kinematics are shown as normalized displacements (b) perpendicular to and (c) along fractures. A gap in the rock temperature time series of location *mh12* ($T_{\mathrm{east}}$) is filled for the time period November 2012 until July 2013 and from August 2014 onwards applying quantile mapping using the best regressors approach (Staub et al., 2016) with a coefficient of determination $R^2 = 0.92$.

fracture kinematics along fracture are either during winter or almost during the whole year. Note that these determined time periods constitute for the further analysis.

## 4.2 Thermo-elastic reversible response and LRM

Figure 7 shows the relation between observed fracture kinematics and rock temperature. Applying the LRM, we obtain the lin-
ear regression coefficients that describe the reversible temperature dependent fracture displacements (black lines in Fig. 7). The fracture displacement at location *mh02* (South, Fig. 5) is almost temperature independent (regression coefficient of $-1.2 \cdot 10^{-2}\,\mathrm{mm/^\circ C}$) except for the winters 2008/2009 and 2014/2015. In contrast, location *mh03* (North, Fig. 5) shows a stronger temperature dependency of $-4.0 \cdot 10^{-2}\,\mathrm{mm/^\circ C}$. At *mh08* (East, Fig. 5), the coefficients are with $-8.3 \cdot 10^{-2}\,\mathrm{mm/^\circ C}$ perpendicular to fracture and $-4.1 \cdot 10^{-2}\,\mathrm{mm/^\circ C}$ along fracture. These temperature dependencies are likely influenced by
the combination of geometric arrangement and acting mechanisms. A potential lack of temperature dependency in the LRM analysis would mean that no reversible or negligible displacements occur that are caused by thermo-elastic strain. Or in other words, irreversible displacements dominate.

Reversible fracture displacement is now modeled for the whole dataset with the *LRM* (see green lines in Fig. 8) using the regression parameters given in Table 2 (light blue shading in Fig. 8). The red line in Fig. 8 represents irreversible displacements,
obtained from subtracting reversible displacement (green line) from the observed displacement (blue line). This analysis clearly

**Table 2.** Regression analysis between temperature (rock or fracture) and observed fracture displacements (perpendicular and along fracture). Regression parameters intercept $\beta_0$ and slope $\beta_1$, correlation coefficient $r$ and coefficient of determination $R^2$ for the time period with the highest correlation coefficient are listed. Depth of the most representative temperature (thermistor $T$) is described in Table 1.

| Location | Temperature (thermistor) | Kinematics | Time period | | $\beta_0$ (mm) | $\beta_1$ (mm/$^\circ$C) | $r$ | $R^2$ |
|---|---|---|---|---|---|---|---|---|
| *mh01* | fracture @ *mh06* ($T2$) | perpendicular | 13 May 2014 | – 22 Jul 2014 | 8.6 | -0.0035 | -0.88 | 0.77 |
| *mh02* | fracture @ *mh04* ($T5$) | perpendicular | 28 Oct 2014 | – 30 Dec 2014 | 19.0 | -0.0127 | -0.96 | 0.92 |
| *mh03* | rock @ *mh12* ($T4$) | perpendicular | 01 Oct 2013 | – 28 Feb 2014 | 43.5 | -0.0404 | -0.96 | 0.92 |
| *mh04* | fracture @ *mh04* ($T4$) | perpendicular | 30 Sep 2014 | – 16 Dec 2014 | 13.4 | -0.0038 | -0.95 | 0.91 |
| *mh06* | rock @ *mh11* ($T4$) | perpendicular | 01 Oct 2013 | – 07 Jan 2014 | 11.2 | -0.0274 | -0.98 | 0.97 |
| *mh06* | fracture @ *mh06* ($T2$) | along | 22 Jul 2014 | – 23 Dec 2014 | -134.0 | -0.0313 | -0.90 | 0.82 |
| *mh08* | rock @ *mh12* ($T4$) | perpendicular | 21 Jan 2014 | – 01 Jul 2014 | 19.8 | -0.0829 | -0.99 | 0.97 |
| *mh08* | rock @ *mh11* ($T4$) | along | 22 Oct 2013 | – 18 Feb 2014 | 43.9 | -0.0407 | -0.95 | 0.91 |
| *mh20* | rock @ *mh11* ($T4$) | perpendicular | 13 May 2014 | – 15 Jul 2014 | 72.2 | -0.1202 | -0.98 | 0.98 |
| *mh20* | rock @ *mh11* ($T4$) | along | 15 Oct 2013 | – 17 Dec 2013 | -19.6 | -0.0696 | -0.98 | 0.96 |
| *mh21* | fracture @ *mh02* ($T6$) | perpendicular | 31 Dec 2013 | – 18 Mar 2014 | 33.0 | -0.0947 | -0.99 | 0.97 |
| *mh21* | rock @ *mh11* ($T4$) | along | 07 Jan 2014 | – 09 Sep 2014 | -127.6 | -0.1620 | -0.99 | 0.97 |
| *mh22* | fracture @ *mh03* ($T4$) | perpendicular | 10 Dec 2013 | – 18 Feb 2014 | 21.3 | -0.0085 | -0.94 | 0.89 |
| *mh22* | rock @ *mh11* ($T4$) | along | 24 Dec 2013 | – 14 Oct 2014 | 81.4 | -0.0363 | -0.97 | 0.93 |

shows that the evolution of irreversible fracture displacement is described for every year by a single phase of quiescence (or solely reversible displacements) followed by a phase of almost linear irreversible displacements once a year. For most locations, including *mh03*, the distinct irreversible phase occurs during the summer, starting when rock temperatures rise above 0$^\circ$C. This likely refers to thawing related processes with melt water that percolates into fractures as a potential cause for this irreversible displacement. At a few locations, such as *mh08*, this linear irreversible phase occurs in autumn when rock temperatures reach freezing conditions, suggesting cryogenic processes (i.e. ice pressure, see Section 1.1) as the causing mechanism. There are however discrepancies to this simple temporal pattern, for example for location *mh03* (see Fig. 8a, black arrows) additional small excursions in displacement occur in summer 2010 and 2015, when summer temperatures are exceptionally high. Although these excursions seem to be reversible, they are not explained by the LRM approach. Furthermore, for location *mh08* in summer, the full amplitude of reversible displacement is not always reproduced by the LRM.

### 4.3 Thawing degree days and summer shift

The summer shift of the fracture kinematics (SHT) and the thawing degree days (TDD) are parameters, allowing to analyze and interpret the inter-annual evolution (Fig. 9). TDD are not computed if the temperature time series contain a gap during summer. A weak correspondence is apparent (see Fig. 14 in appendix A) for locations with aspects to the north and east. This hints on a substantial influence of rock temperature and therefore incoming conductive energy fluxes. Interestingly, at locations exposed

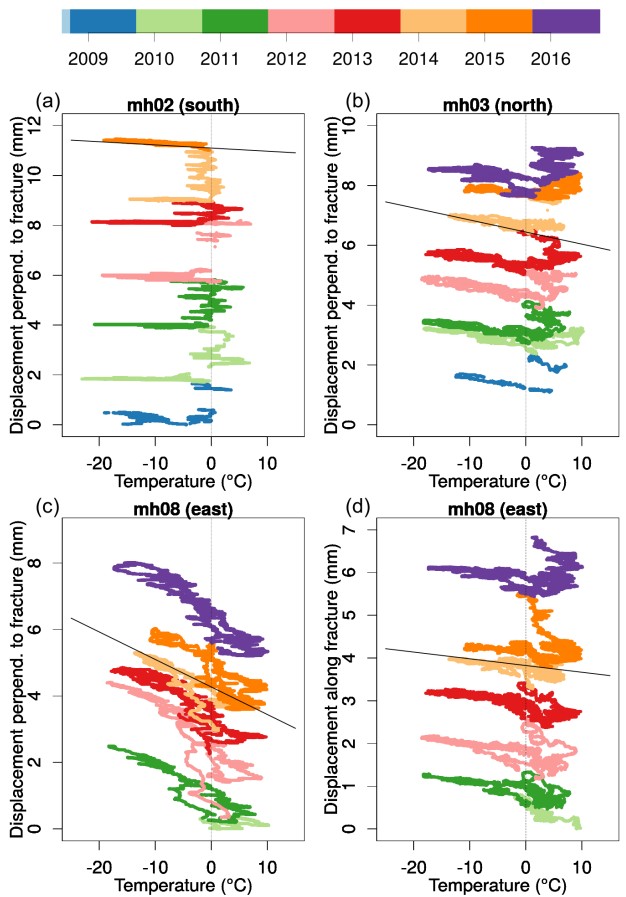

**Figure 7.** Temperature dependency of fracture displacements for location *mh02* (perpendicular to fracture), *mh03* (perpendicular to fracture) and *mh08* (perpendicular to and along fracture). Discrete colors indicate hydrological years (1 October – 30 September). Black lines indicate the linear regression function determined by the regression analysis (see Table 2).

to the south, SHT seems independent of TDD. The local break-off at location *mh02* occurred in summer 2015 (described in first paragraph of Section 4, page 11). This summer exhibits a record high in TDD at all locations.

## 4.4   Irreversibility index

The irreversibility index indicates the onset of irreversible displacement and is shown in Fig. 10 for displacements perpendicular to fractures. In general, this index shows once a year a period with sudden increases of irreversible displacement at all locations. High index values can be observed in summer (positive temperatures) at location *mh02* (South) and *mh03* (North), during thawing period, while in winter low indices occur without any peaks (see Fig. 10a and 10b). The irreversibility index shows

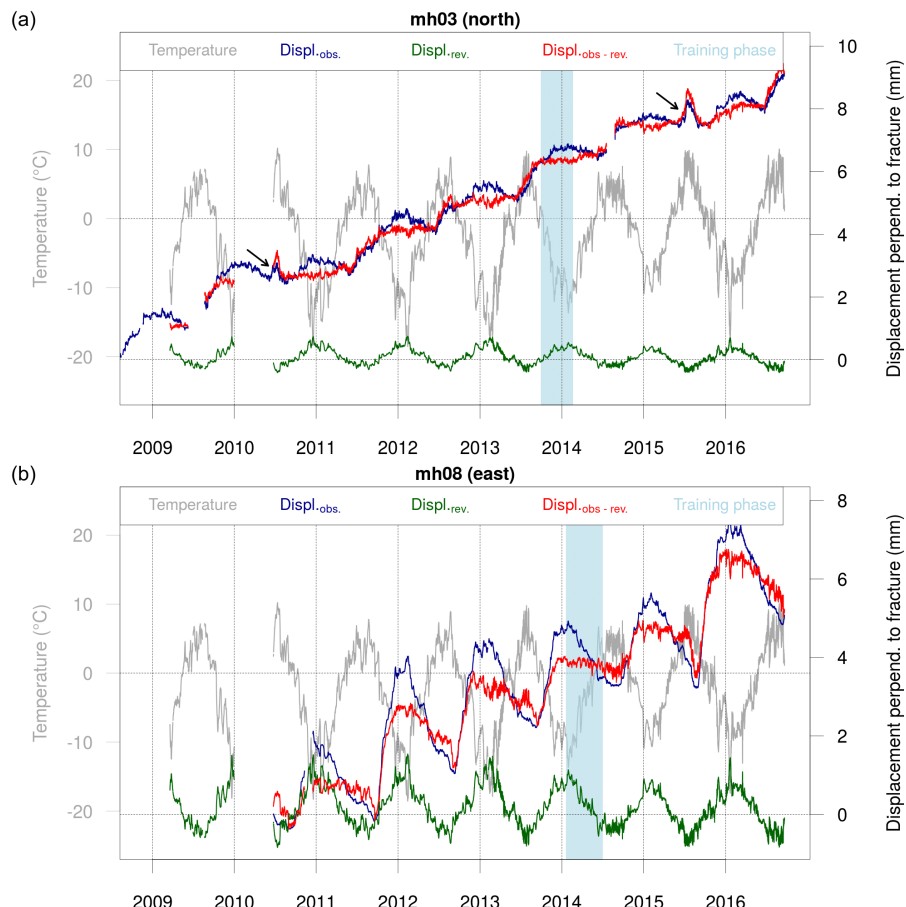

**Figure 8.** LRM (green) applied to the observed displacements (blue) perpendicular to the fracture at location *mh03* (a) and *mh08* (b). The reversible component (green) due to thermo-elastic strains in rock can be modeled by a linear regression model (LRM) with temperature (dark gray) and displacement measurements during a training period of several months (light blue shading) as input data. Subtracting these reversible displacements from the observed data results in the red line, referred to as irreversible fracture displacement.

that irreversible displacement is related to positive temperatures, which further supports our findings from the relation between SHT and TDD (Fig. 9).

In contrast, for location *mh08* a high irreversibility occurs in autumn when temperatures drop below $0°$ C, suggesting freezing as a dominant process. Note, these periods of high indices correspond to the irreversible displacement phase obtained from the LRM.

The reversible excursions from the LRM at location *mh03* in summer 2010 and 2015 are picked up by increased indices. However, they are reversible displacements that are not represented by the LRM. This points to a potential additional reversible process that cannot be explained only by the thermo-elastic strain.

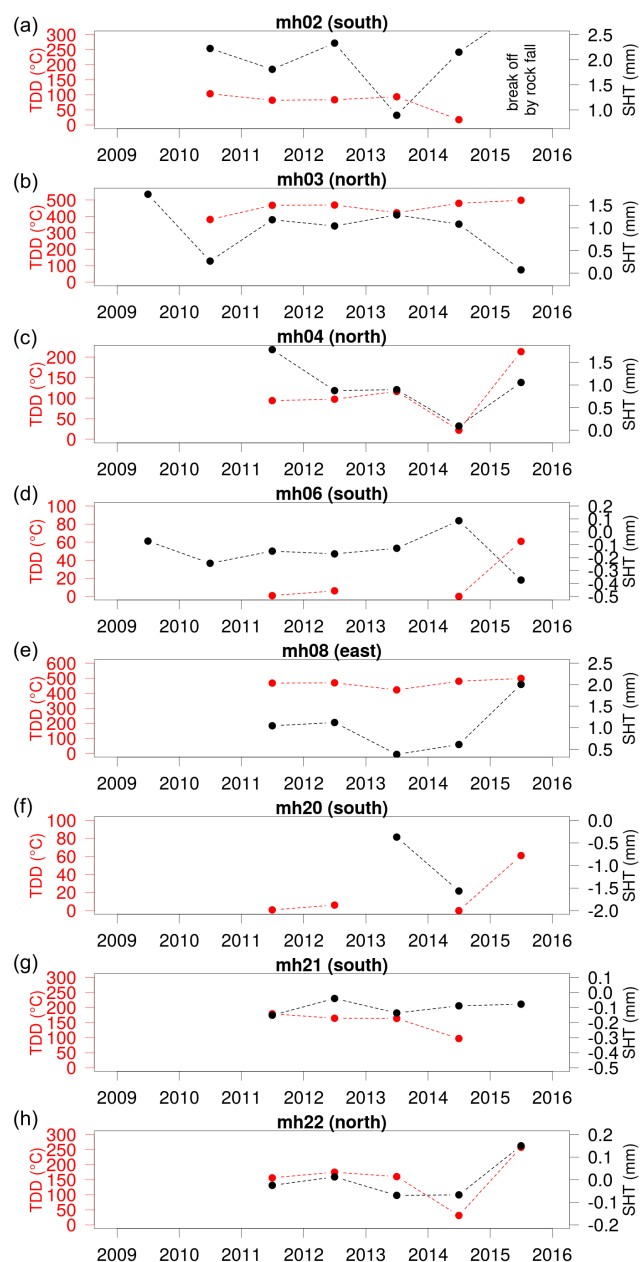

**Figure 9.** Inter-annual variability of thawing degree days (TDD) and summer shift of fracture kinematics (SHT) perpendicular to fractures for all locations. Data at location *mh02* is missing from 2015 onwards due to the break-off and the TDD values at a few locations for the year 2014 are removed due to missing or incomplete temperature data.

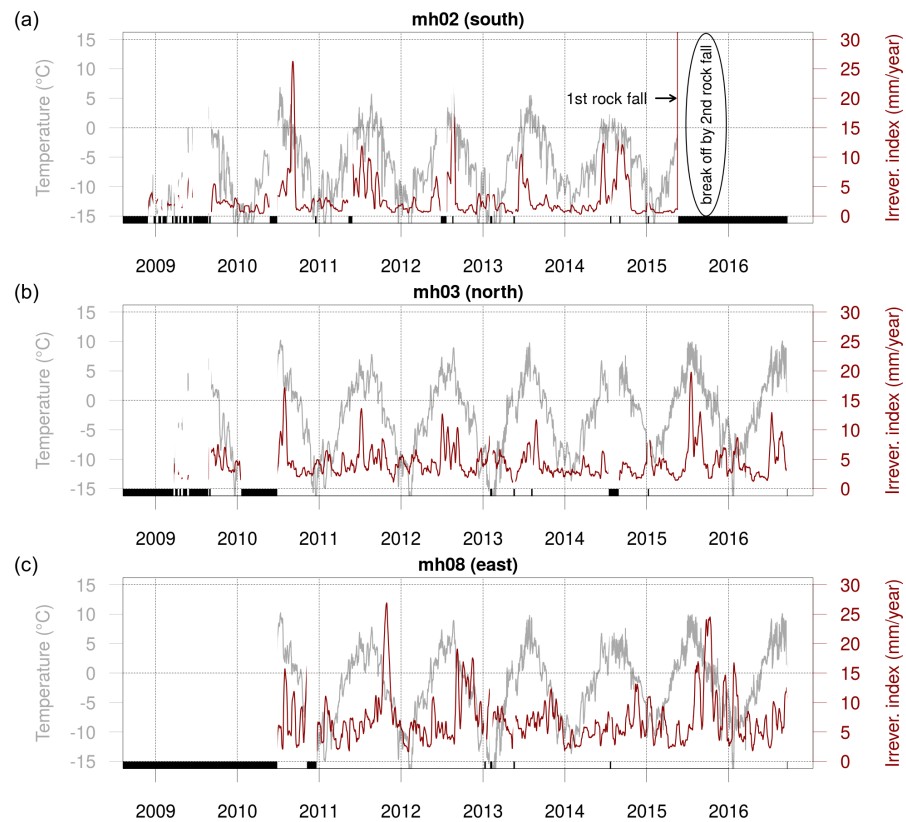

**Figure 10.** Irreversibility index for (a) location *mh02* (south), (b) location *mh03* (North) and (c) location *mh08* (East, on ridge) as an indicator for periods, where the irreversible displacement dominates. Black bars indicate periods where no or reduced data is available.

## 5 Discussion

This study aims at quantifying and separating reversible and in particular irreversible fracture kinematics in relation with environmental forcing. The main processes leading to fracture kinematics are presented in Fig. 1, enabling to isolate different processes from the field observations. Possible interactions between the different processes are not considered but may well 5 occur in nature. Using our quantitative approach, we are able (i) to separate reversible from irreversible fracture kinematics and (ii) to produce a new irreversibility index. This new metric provides a useful indication for the occurrence and timing of irreversible displacement and thereby contributes towards rock slope stability assessment. In the following, we discuss the research questions formulated in Section 1.2.

### 5.1 Separation of the reversible fracture kinematics

10 Very high coefficients of determination given by the regression analysis (see Table 2) support the suggested linear relation between temperature and fracture kinematics (see Fig. 1b). The regression analysis is only based on few assumptions (see

Section 3.1), thus preventing coincidental relations. The duration of the training periods (set to a minimum of 70 days) prevent such high coefficients caused by an irreversible process. As the best coefficients are obtained in winter, reversible thermo-elastic strain dominates during this period. It further supports the postulated existence of intra-annual periods with negligible irreversible displacements. Temperatures deeper in rock/fracture might cause even higher correlation coefficients, as the correlation coefficient mostly increases with increasing depth of the temperature measurement. But it is difficult to estimate a representative depth for temperature measurements as the temperature variations are attenuated with increasing depth and the deepest available rock temperature measurement on Matterhorn is at 0.85 m depth.

The linear regression model (LRM) can reliably reproduce the thermo-elastic strain for a given temperature, and therefore can be used to describe the observed reversible displacement component in all instrumented fractures. Furthermore our analysis shows that a selected single time period of a few months is representative for the reversible component in displacement for the whole time series when the process thermo-elastic strain strongly dominates (e.g. winter). Therefore, such a quiescent time period can be used as the training phase for the LRM. The exception is at location *mh02* (see Fig. 12) where the reversible fracture displacement is almost negligible apart from winter 2014/2015 after which the small failure occurred. This location further shows an annually changing relation between fracture displacement and temperature (see Fig. 7), which is an exception in this data set. Otherwise, the amplitude of reversible displacements varies strongly from location to location. Although we expect the thermal expansion coefficient of pure rock material to be very similar, we explain this variation by highly variable volume or length of rock wall material influencing an individual fracture and by the spatial heterogeneity in thermal conditions at depth. Hence, the magnitude of the reversible fracture displacement, caused by thermo-elastic strain, is influenced by the individual geometric mesoscale arrangement of each fracture.

In principle, LRM can be applied the same way to fracture kinematics perpendicular to and along fracture (see Fig. 13 in appendix A). But the kinematics along fracture is much more sensitive to the geometric mesoscale arrangement of the fracture. Assuming for instance the rock masses aside the fracture have the same size and thermal condition, the thermo-elastic strain is also the same and no relative displacement along fracture is measured.

Observed reversible excursions in displacement at location *mh03* in summer 2010 as well as in summer 2015 are not caused by thermo-elastic stress which is also evident from the high values of the irreversibility index (Fig. 10). These excursions in displacement may be caused by a non-local effect or points to an additional unidentified process causing reversible displacement. These excursions sporadically occur during summer with very high temperatures. Ice pressure and its release by melting can also produce reversible excursions with a fracture opening during freezing and a fracture closing during melting. However, the closing phase would have to start at the melt onset, which is clearly not observed. Thus ice formation is not playing a dominant role for reversible fracture kinematics.

## 5.2 Inter-annual pattern of irreversible fracture kinematics

Close to a decade of field measurement provides enough data for inter-annual analysis of fracture kinematics. In general, all instrumented locations show a trend of fracture opening or closing perpendicular to fractures, but with different rates. At each individual location, the temporal pattern of displacements is very similar every year, but the irreversible summer shift (SHT)

slightly varies over time. According to our analysis, this summer shift seems at least for north facing locations to correlate slightly with an increasing total amount of available energy (TDD). This suggests that further warming and therefore increasing TDD's cause thawing of permafrost at greater depth, potentially leading to an increase in summer shifts (SHT). Percolating water allows effective heat transport along fractures leading to faster temperature increase in fractured rock mass than in intact rock. Additionally, water percolation can affect the shear resistance along fractures and lead to a decrease in friction, which can cause irreversible displacement. For example at location *mh02*, enhanced availability of water from snow melt after summer snowfall events seems to cause accelerated irreversible displacements.

As TDD derived from mean daily rock temperature, relation between summer shift and TDD in south exposed and warmer rock should be interpreted carefully. Rapid variation of temperature with short peaks above $0°$ C can lead to thawing even when the mean daily temperature stays below $0°$ C. This is often the case at locations exposed to strong solar radiation (south facing), even at winter time, and might explain why the TDD at the south exposed locations do not correlate with the summer shift (e.g. *mh02* or *mh21*).

The presented summer shift only provides total displacement between two winters without any intra-annual information. In contrast, the irreversibility index can be seen as a proxy of impending rockfall activity and reveals information on the short term evolution of the irreversible fracture kinematics all year round, even if the total summer shift (SHT) is small. Despite based on local measurements, such an index can help to identify periods of enhanced irreversible fracture kinematics or risk for failure (see Fig. 2). For example, a strong increase was observed in early summer 2015 at location *mh02*, followed by several small rockfalls and a final break-off (approx. $2 - 3\,\text{m}^3$, timing indicated in Fig. 10a). Similar at location *mh03*, irreversible displacement occurs during the melt period, which is likely related to a reduction of friction along a fracture line.

However, there are also irreversibility index peaks in autumn, e.g. at location *mh08* (East, on ridge, Fig. 10c), which do not correlate with thawing days but with rapid cooling and freezing in autumn. In this case, the growth of ice in late autumn acts as a driving factor through increasing ice pressure by cryogenic processes. Interestingly no fracture closing is observed during ice melt period in the subsequent summer indicating irreversibility of such a process. Such thermo-elastic and cryogenic forcing of fracture kinematics has been hypothesized by Hasler et al. (2012), but their data was not fully conclusive on this point due to the short duration of the data set (1–2 years).

## 5.3 Environmental controlling of irreversible fracture kinematics

Combined analysis of LRM and irreversibility index exhibits distinct periods of solely reversible fracture kinematics and others with additional irreversible fracture kinematics. Irreversible displacement seems to be driven by environmental conditions, namely by rock temperature above $0°$ C (indicating thawing) or less commonly by periods of freezing conditions. In the main winter time (temperatures well below freezing) after the initial cooling phase, none of the instrumented fractures shows irreversible displacement. Seasonal freezing and thawing of the rock mass in the active layer can influence fracture kinematics in several ways and can lead to irreversible displacements. On the one hand warming influences the fracture toughness of rock bridges, creep of ice and total friction along existing shear zones (Krautblatter et al., 2013). On the other hand, water from the surface mainly by snow melt can percolate into fractures. This increased water availability can refreeze at the permafrost

table and cause cryogenic pressure. If the water and/or heat supply is high enough, the water column can rise and enhance hydro pressure. But high water columns are rather unlikely at the Matterhorn field site, because it is located on the ridge with steep, laterally open fractures. Therefore, the suggested patterns for cryogenic and hydrostatic processes in Fig. 1b cannot be confirmed. These patterns may be oversimplified, as this study shows that the related processes are often superimposed on and not clearly distinguishable.

## 6   Conclusions

Knowledge of processes and factors affecting rock slope stability is essential for detecting and monitoring potentially hazardous rock slopes. A unique eight year time series of fracture kinematics is presented, providing new insights on fracture kinematics with respect to thermal conditions on steep high-alpine rock slopes. The intra- and inter-annual behavior of the fracture kinematics strongly varies between locations, but patterns at individual locations are consistent over the entire observation period. Longterm monitoring at multiple fractures thus essentially helps to improve the process understanding of fracture kinematics.

The regression analysis highlights periods with a significant negative correlation between fracture kinematics perpendicular to fracture and temperature for all locations. Interestingly, the most representative time periods used for training the LRM occur in winter and early spring. The proposed LRM approach provides a tool for systematic analysis of fracture kinematics and was successful in separating reversible from irreversible displacements. An irreversibility index was built to detect irreversible displacement and its link to environmental forcing. Eight years of relative surface displacement measurements show that reversible fracture kinematics caused by thermo-elastic strains of the material is occurring at all locations except one all year round, but are temporarily superimposed by other processes. During additional phases of irreversible displacement with a stepwise behavior occur mostly during periods with temperature above $0°$ C suggesting a decrease in friction along fractures as a responsible process. At one location, ice formation due to freezing during the onset of the winter also causes irreversible displacements. These results are supported by the developed irreversibility index. As irreversibility can lead to rock slope failure, quantifying irreversible kinematics is a first step toward assessing rock slope stability.

However, this approach to measure relative surface displacement has limited time resolution and provides only point information from near the surface and with a limited spatial coverage. Ongoing analysis of micro-seismic activity, currently measured on this field site, could potentially give insights with a very high temporal resolution and a extended spatial coverage. Clustering of micro-seismic events coincident to rock fracturing (Murton et al., 2016) could reveal insights into relevant fracturing types. Coupling spatio-temporal characterization of irreversible displacement with internal progression of microcrack activity could significantly improve process understanding and be applied in the context of early warning system.

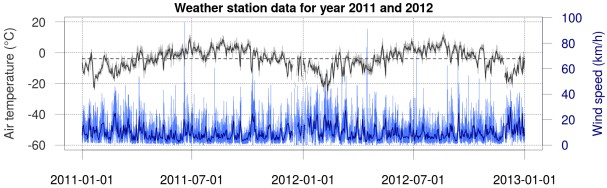

**Figure 11.** Time series of the in situ installed Vaisala WXT520 weather station providing air temperature and wind speed for the years 2011 and 2012. 10 minutes averages are shown in gray (air temperature) and lightblue (wind speed) whereas weekly averages are shown in darkgray (air temperature) and darkblue (wind speed). Dashed darkgray line represents the mean temperature.

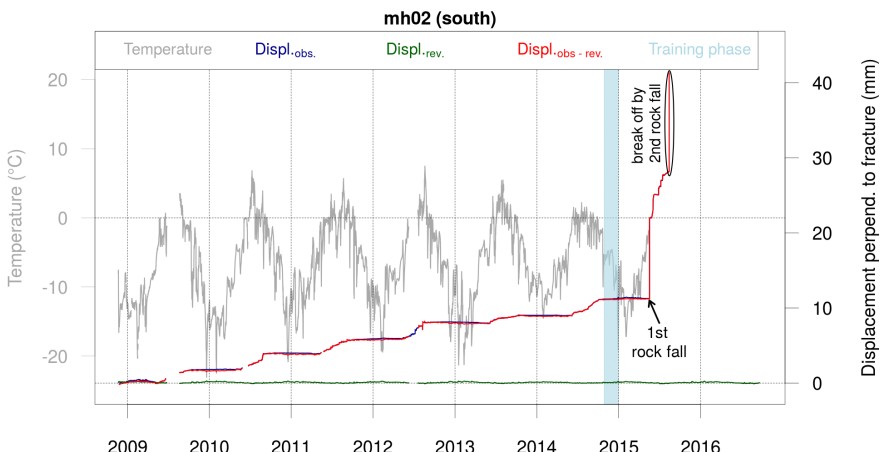

**Figure 12.** LRM (green) applied to the observed displacements (blue) perpendicular to the fracture at location *mh02*. The reversible component (green) due to thermo-elastic strains in rock can be modeled by a linear regression model (LRM) with temperature as input data (dark gray) and displacement measurements during a training period of several months (light blue shading). Subtracting these reversible displacements from the observed data results in the red line, referred to as irreversible fracture displacement.

## Appendix A: Supplementary figures

## Appendix B: Data availability

All used data (processed and aggregated as 10 min averages) is available in the supplementary as csv-file for each location. The meta information is given in Table 1 on page 9. Additional data can be accessed via the PermaSense GSN data portal (data.permasense.ch). A system documentation and tutorial for online data access is available on the PermaSense project web page (www.permasense.ch/data-access/permasense-data.html).

*Author contributions.* Jan Beutel and Andreas Hasler designed the field experiment and installed the sensors in 2010 and 2012. Jan Beutel and Samuel Weber have done maintenance work and data management tasks since spring 2012. The analysis code in R was written by

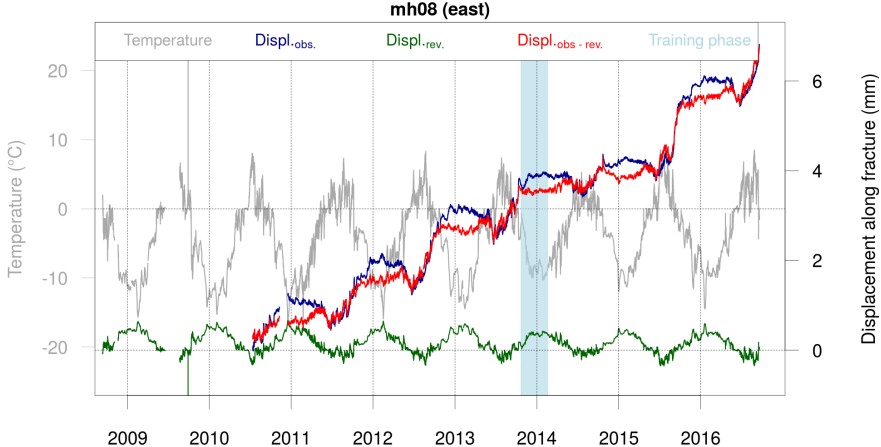

**Figure 13.** LRM (green) applied to the observed displacements (blue) along the fracture at location *mh08*. The reversible component (green) due to thermo-elastic strains in rock can be modeled by a linear regression model (LRM) with temperature as input data (dark gray) and displacement measurements during a training period of several months (light blue shading). Subtracting these reversible displacements from the observed data results in the red line, referred to as irreversible fracture displacement.

Andreas Hasler and Samuel Weber. Samuel Weber developed the model code as well as the irreversibility index and performed the data analysis. Samuel Weber prepared the manuscript with substantial contribution of all co-authors.

*Acknowledgements.* The work presented in this manuscript is part of the project X-Sense 2 and was financed by nano-tera.ch (Ref.: 530659). We acknowledge the PermaSense team, namely Tonio Gsell and Christoph Walser, who provided valuable support with the development of
5    measurement devices, in the field and with data management. We thank Max Maisch for providing us geomorphic considerations for Fig. 3 and Marcia Phillips for editing the English. Reviews from Valentin Gischig and three anonymous referees provided valuable comments that helped to improve the manuscript substantially.

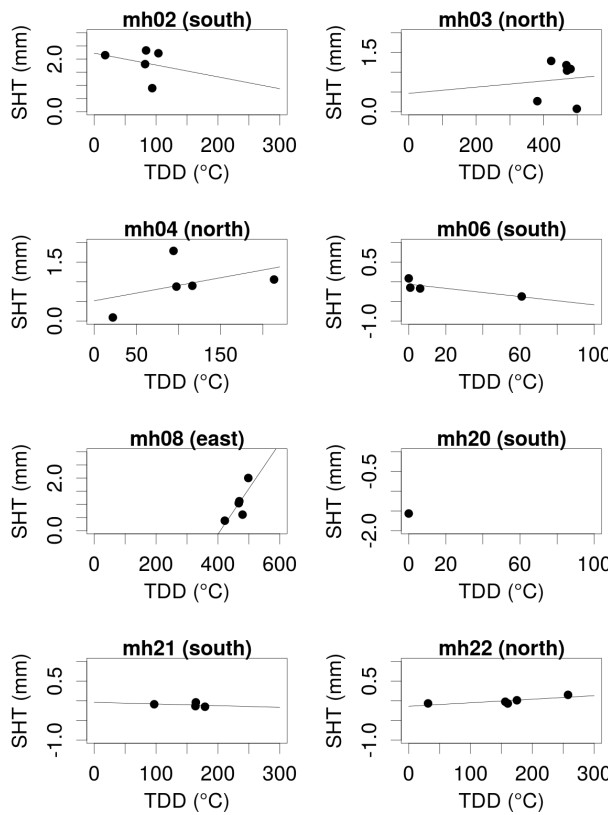

**Figure 14.** Summer shift (SHT$_{summer}$) of displacement perpendicular to fracture against yearly thawing degree days (TDD$_{year}$) for locations *mh02*, *mh03*, *mh04*, *mh08*, *mh21* and *mh22*. The black line indicates the regression function.

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
