# Peer review of "Quantifying irreversible movement in steep fractured bedrock permafrost on Matterhorn (CH)"

_The Cryosphere, 2016_

## Referee Comment (RC1) · Anonymous Referee #1 · 17 Jul 2016

This paper presents seven years of rock-fracture movements from nine sites at Matterhorn, proposes a technique separating the total movements into several components. Both data and analysis are new and valuable. I recommend acceptance after minor modifications.

General comments: 1. Analysis methods are in places not understandable: e.g. Section 4.2. Some newly defined terms (e.g. OFST, LRM+) also do not seem to represent the actual features, so they are difficult to understand. 2. Why temperature is represented by data at 85 cm depth (instead of surface T)? Does the value best correlate with deformation? If it is true, why?

Specific comments: P1 L8 Insert 'that' after assuming. P2 L32 Does 'thermal expansion' here mean D2 or include also D3? Ice pressure and its release by melting can

also produce reversible movement. P3 L4 Insert 'us' after allows. P6 L31 . . .strong wind results in a preferential snow deposition in fractures. . . (Insert 's' and delete comma.) P7 L4 Does 'along fracture' mean that the crackmeter does not cross a fracture? A photo or illustration of the installation will be helpful. P8 Fig.3 caption Scap of the 2003 rockfall. . . P8 L9 Why did you use T at 85 cm depth? P9 L3 Where is 'ti' in Eq. 2? P10 L8 Insert 'of' after 'approximation'? P10 L16 What does 'date(Trock < −1 C)|May 1' mean? P10 L27-29 Should this be described in Section 4.4 (equation 6)? P11 L19 Perhaps 'installation' could be deleted. P12 Fig. 5 caption L2: I suggest to insert (a) after 'at 85 cm depth'. L3: I suggest '. . . represented by (b) perpendicular to and (c) along fracture'. P13 Fig. 6 Why deformation does not start from zero (cf. Fig. 5). P13 Fig. 6 caption Replace 'parallel' by 'along'? P13 L4-5 Note that. . .: I cannot understand this sentence. P14 L3 Insert comma after 'Assuming'. P14 L13-14 Hinting on. . .: Also not understandable. Lacking a verb. P14 L17 Replace 'Section 5' by 'Figure 5'? P15 Fig. 7 Why deformation does not start from zero? P18 L8-9 Which data do show 'this summer offset seems to correlate slightly with an increasing total amount of energy available (TDD)'? P18 L32 What is 'the hypothesis of Hasler et al. (2012)'? P19 L7-8 'the water can easily drain through the strongly fractured rock and the water availability is limited': Does this situation fit any places in the rockwall? Aren't there any locations topographically favorable for water storage?

---

## Referee Comment (RC2) · Anonymous Referee #2 · 22 Aug 2016

GENERAL COMMENTS

The authors propose an empirical/statistical model aimed at separating reversible components of fracture deformation, due to thermo-elastic strains in alpine high-elevation permafrost environments, and the irreversible (plastic) component due to other processes. The topic is interesting and of interest for The Cryosphere. The work is based on a very interesting 7-year time series of fracture displacements recorded at several locations at the Matterhorn (Switzerland) by a monitoring network set up by Hasler et al (2012). Nevertheless, my review pointed out a number of serious scientific issues, which are listed in the following general comments and in the following "Detailed comments" section. I suggest that these points must be carefully addressed before the manuscript can be published in a high-level journal as The Cryosphere.

1) The abstract is quite too long and should be more focused and to-the-point;

2) English could be generally improved using shorter and more focused sentences;

3) The "mechanical conceptual model" (Section 2) is characterized by some weaknesses and is not used later in the paper, which then focus on an empirical/statistical model. The authors seem to want to add a "rock mechanics taste" to the work, but tend to mix some different concepts and quantities and use terms as "fracture dynamics" which sound ambiguous to people from the geological and engineering rock mechanics communities (see detailed comments);

4) The empirical/statistical model, making the core of the work, is biased by strong assumptions leading to somehow obvious results and poor predictive capability (see different detailed comments below). Actually, it is difficult for me to see either the scientific advance or the practical contribution of this work. In fact, the statistical model aims at discriminating thermo-mechanical elastic displacements, which are indeed small and of the same order of magnitude of possible precursors of rock slope instability (the latter can also follow very different patterns). This seems to suggest that the reliability of the method is low for small irreversible displacements and useless when irreversible displacements become larger. Finally, irreversible displacements are not investigated themselves thus the method cannot be used to predict rock slope failures (as promised in the abstract)

5) The most interesting contribution seen here is monitoring, providing a continuous, 7-years long time series of displacements. Nevertheless, this contribution originates from the previous work by Hasler et al 2012.

DETAILED COMMENTS

Page 1, line 4: (and elsewhere in the manuscript): "fracture dynamics" is a confusing term to members of the rock mechanics communities (both geoscience and engineering): in fact the term "dynamics" usually refer to fracture mechanics (micro- or meso-

modes of failure and related mechanical models and parameters; see e.g. Paterson & Wong) under dynamic loading conditions. Instead, the author simply refer to the temporal pattern of movements along or perpendicular to fractures. Why not use a simple term as "fracture kinematics"?

Page 1, line 9: "gravity-driven slope failure": Rock slope failure? Landslide?

Page 1, lines 12-13: "enables a local assessment of rock wall stability": actually, the presented work just aims at depurate a time series of displacement along fractures from the elastic thermal component. No analysis of the spatial-temporal patterns, mechanisms and triggers of irreversible displacements is proposed, thus I do not understand how rock wall stability is dealt with here.

Page 2, line 4: "frozen rock masses": the authors focus on rock masses with ice-filled discontinuities and exclude ice-free frozen rocks, where a thermal elastic strains indeed occur. This is ok, but I suggest that this should be declared clearly as an assumption at the beginning of the analysis, also suggesting the expected differences in the behaviors of ice-filled and ice-free rock masses with respect to slope instability. This would be very useful to non-permafrost-experts involved in the analysis of slope instability at high altitudes.

Page 2, line 21: "Intact high prosity rocks": and what about low porosity rocks, which form most of the Alps?

Page 3, line 25 "thermo-elastic induced strains": the conceptual model of the authors is based on a balance of driving and resisting forces. Strains are not forces, but are related to forces by a specified rheology and geometry (i.e. Stress distribution). Balancing the contribution of strains is formally incorrect, although this has no consequence on the analysis because the mechanical model is actually not used in the following (but it is another weakness of this work; see General Comments)

Page 3, line 27: "creep and fracture of ice": here the authors include among resisting

forces some processes and quantities that are not forces. Creep is a time dependent deformation of materials, including a large variety of physical processes at micro to macro scales. Fracture is brittle failure of solids. I understand that ice deformation and failure reduces stresses through plastic work, but again it is formally not correct to include these processes as forces.

Page 3, line 27: "fracture infill": strength of fracture infill?

Page 3, line 31: "reversible and irreversible": elastic and plastic?

Page 3, line 33 and Page 4, line 1: I am not convinced about the physical consistency of the "temperature-fracture deformation relationships". It is well known from a huge laboratory rock mechanics literature that the rheology (stress-strain relationships, brittle vs ductile behavior) of rocks depends on temperature. Thus, it would not be possible in principle to define unique temperature-strain relationships, especially when dealing with creep, which is non-linear and time dependent even at constant temperature. I understand that authors just refer to individual existing fracture deformations and guess that they assume linear elastic-perfectly plastic rheology in the considered temperature range. Nevertheless, the authors should clearly state and support their assumptions and related limitations: are they sure that stress-strain-temperature relationships for ice filled fractures (and even more for fractured rock masses!) are as simple as they state? Are they able to provide experimental data or literature to support that?

Page 6, line 23: "these statements are validated": in the following, the authors switch from a conceptual mechanical model to a simplified statistical one to discriminate reversible and irreversible movements along monitored fractures. Nevertheless, the postulated origins of irreversible movements, I.e. "Cryogenic" in winter and "hydro" in summer, although reasonable, are not validated by data and analysis. No information is provided about the state of ice filling in fractures, and there is no correlation between hydrological parameters (e.g. Rainfall) and irreversible movements.

Page 6, line 29: "heterogeneous": in which sense?

Page 6, line 30: rainfall, cold winter temperature, exposure etc.: please provide quantitative values/ranges typical of the studied environments.

Page 7, line 5: could the authors explain why they measured temperature down to 85cm and not deeper? This also applies elsewhere in the manuscript. Which are the other measuring depths, and why temperature profiles are not used / presented?

Page 7, lines 5-6: "high resolution images": what are these used for, also considering that pixel resolution is of the same order of magnitude of the fracture displacements recorded in seven years?

Page 7, line 13: "aggregated": cumulated or averaged?

Page 8, line 4: "Staub et al": manuscripts in review are not citable.

Page 8, line 9: "Used temperature ....at 85cm depth": why?

Page 9, lines 3-10: the statistical linear model for the reversible deformations is poorly explained and supported: does it apply at the same way to shear and normal fracture displacements? How is the data population related to reversible movements separated from the irreversible movements occurring in winter for fitting purposes? Which are the best-fit statistical parameters of the model and related measures of statistical performance? These are not reported and the reader is forced to believe that the model is robust. This is a major scientific weakness of the work and the authors should work more on this.

Page 9, line 14: 28 days window length: one month is a long smoothing period, could the authors explain why they used such a long time interval? In general, one could expect that excessive smoothing may "kill" some important signals on shorter timescales.

Page 10, line 11: "due to creeping": this part is obscure and, again, I cannot understand how the authors are able to separate the population of reversible vs. irreversible winter deformations.

[Figure]

Page 10, line 21: "in winter. . ...we assume that deformation by the thermos-mechanical induced strain dominated": this indeed remains a strong assumption, possibly significantly biasing the model. The authors should try to support this better.

Page 11, Figure 4 (and related text): the piecewise linear regression model sounds over-simplified and biased by different strong assumptions including the following: 1) winter deformation is always reversible (or, at least, the same reversible deformation fitted in an early "training period" occur every winter – this may be not true as the rock mass accumulates damage); 2) the beginning of the "creeping" phases can be pre-defined and is the same every year; 3) displacement time series in the creeping phase are linear. I suggest that these assumptions pose too many constraints on the model and hampers its application to prediction/forecasting, except in very simple cases.

Page 11, line 23: ". . .a field site can not be described by a single measurement location. . ..": this seems quite obvious, and things are even worse when dealing with rock masses instead of individual fractures.

Page 12, lines 9-10: "indicated by a black line in Figure 6": this is unclear or incorrect. The black lines seem linear regression functions, not their coefficients (which are never reported in the paper; instead, the authors should provide tables of best-fitting function parameters and regression quality statistics or indices to demonstrate the performance of their statistical model). Moreover, it is difficult to understand why the black lines are plotted at these positions (why don't they intersect the x-axis in zero? What is actually fited?)

Page 13, lines 4-5: "note that. . ...deformation": incomplete statement.

Page 13, lines 5-6: "reduced data input": the authors' approach is to fit very limited time windows and then extrapolate the results. But in this way, they are not able to obtain a model fitting the entire dataset, which is particularly important to empirically fit time-dependent movements (creep). Also, in this way the potential of the beautiful 7-year presented monitoring series is not exploited.

Page 14, line 1: "this likely indicates thawing related processes": this is obviously reasonable, but but still unsupported by specific analyses. "assuming that water is available. . ...deformation": same comment.

Page 17, lines 4-5: "one single. . .. . .fracture deformation": the result is reasonable in some specific conditions (individual fracture displacements vs. rock mass, low strain, low damage, simple failure kinematics causing block movements), but is biased by the strong assumptions on which the model is based (what is reversible or irreversible deformation?)

Page 18, section 6.2: a qualitative analysis of raw data would have brought the same observations / conclusions, suggesting that the data (following the work of Hasler et al 2012) are very interesting, but the proposed model does not bring significant contributions or advantages (especially in a predictive perspective)

---

## Author Comment (AC1) · 30 Sep 2016

Dear Mr. Isaksen and anonymous Referees,

We would like to thank for the detailed comments and constructive suggestions, which helped us to improve the manuscript. We hope that we have adequately addressed and answered all reviewer comments and changed the manuscript accordingly.

The referees state the topic is interesting and based on a very interesting 7 year time series of fracture displacements recorded at several locations at the Matterhorn. They highlighted several concerns which mainly concerned the clarity of the methodology, the focus and main result of the study and the introductory background information.

We addressed all these issues (and all the specific ones) raised by the referees and

briefly outline here the more substantial changes/ revisions below.

- Regarding the misinterpretation of the focus: The focus of this study is not to predicting rock slope instabilities. To address this, we clarified the focus, purpose and novelty of this study in the abstract and introduction. Regarding the weakness of the initial conceptual model: We agree, the initial conceptual model was not consistent and contained some weaknesses. To be more precise, we now use the term "fracture kinematics" instead of "fracture dynamics". We rewrote and shortened the conceptual model to an overview of the processes and related environmental controls and clarified the aim and research questions of the study in a separate section.

- Regarding difficulty in understanding methodology: We simplified and clarified the methods. We revised and clarified the whole method section. In particular, the LRM+ model was removed. Although it reproduced quite well fracture kinematics, it was not crucial for the main focus and analysis of this manuscript and could confuse readers. We also changed the term "summer offset" to "summer shift" with the abbreviation "SHT". We further extended and improved the regression analysis to investigate the relation between fracture kinematics and temperature.

- Regarding the criticism of referee 2 that a qualitative analysis of raw data would have brought the same observations/conclusions, but the proposed model does not bring significant contributions or advantages: We disagree on this point. This work provides a new quantitative analysis based on a significantly longer time series (7 years vs. 2 years). The scientific advance of this contribution is to distinguish phases as well as the timing of irreversible displacements. Timing of irreversible kinematics is crucial to link the acting mechanisms to environmental forcing. Furthermore, the developed irreversible index provides useful indication on rock wall stability.

In the revised manuscript we addressed all the reviewers' comments and added in the general response one by one explanations and comments to the specific points of the referees. We also added additional figures and changed the figures in the manuscript according to the comments.

With kind regards

Samuel Weber
On behalf of all authors

Please also note the supplement to this comment:
http://www.the-cryosphere-discuss.net/tc-2016-136/tc-2016-136-AC1-supplement.pdf

**Supplement:**

**Reply to comments made by Anonymous Referee #1 (doi:10.5194/tc-2016-136-RC1).**
We thank Anonymous Referee #1 for its review and suggestions for improvement. Referee comments indicated as "RC:", author reply as "AR:". Only sections requiring a reply are reproduced.

RC: GENERAL COMMENT 1. Analysis methods are in places not understandable: e.g. Section 4.2. Some newly defined terms (e.g. OFST, LRM+) also do not seem to represent the actual features, so they are difficult to understand.
AR: We revised and clarified the whole method section. In particular, the LRM+ model was removed. Although it reproduced quite well fracture kinematics, it was not crucial for the main focus and analysis of this manuscript and could confuse readers. We changed the term "summer offset" to "summer shift" with the abbreviation "SHT". An improved explanation of this shift is given on page 10, line 25.

RC: GENERAL COMMENT 2. Why temperature is represented by data at 85 cm depth (instead of surface T)? Does the value best correlate with deformation? If it is true, why?
AR: Surface temperature is strongly influenced by daily fluctuations. The temperature is strongly attenuated with depth and is a more suitable representation of a seasonal signal. We added a table (Table 1 in manuscript on page 9) with an overview of all available temperature measurements (rock temperature at different depths and temperature in fractures). In the revised manuscript, we applied a best fit analysis using all available rock and fracture temperature data. With this we determined the most representative temperature measurement for modeling the reversible thermo-mechanically induced fracture kinematics. The best trainings periods are shown in Table 2 on page 13.

RC: P1 L8 Insert 'that' after assuming.
AR: Done.

RC: P2 L32 Does 'thermal expansion' here mean D2 or include also D3? Ice pressure and its release by melting can also produce reversible movement.
AR: We agree, the initial conceptual model was not consistent and contained some weaknesses. To be more precise, we now use the term "fracture kinematics" instead of "fracture dynamics". We refocused the the conceptual model to an overview of the processes and related environmental controls and clarified the aim and research questions of the study in a separate section. We also clarified in the manuscript that ice pressure and its release by melting can also produce reversible fracture kinematics.

RC: P3 L4 Insert 'us' after allows.
AR: Done.

RC: P6 L31 . . .strong wind results in a preferential snow deposition in fractures. . . (Insert 's' and delete comma.)
AR: Done.

RC: P7 L4 Does 'along fracture' mean that the crackmeter does not cross a fracture? A photo or illustration of the installation will be helpful.
AR: We adapted Figure 3 (new Figure 5 on page 8) and added a photo with a sketch that illustrates locations instrumented with two crackmeters.

RC: P8 Fig.3 caption Scap of the 2003 rockfall.
AR: Done.

RC: P8 L9 Why did you use T at 85 cm depth?
AR: This point was addressed in detail in GENERAL COMMENT 2.

RC: P9 L3 Where is 'ti' in Eq. 2?
AR: This was a mistake. We have removed it.

RC: P10 L8 Insert 'of' after 'approximation'?
AR: Done.

RC: P10 L16 What does 'date(Trock < −1 C)|May 1' mean?
AR: This issue was addressed and clarified by a more detailed explanation on page 11, lines 1-3.

RC: P10 L27-29 Should this be described in Section 4.4 (equation 6)?
AR: This is correct and addressed in the previous referee comment RC: P10 L16.

RC: P11 L19 Perhaps 'installation' could be deleted.
AR: Done.

RC: P12 Fig. 5 caption L2: I suggest to insert (a) after 'at 85 cm depth'.
AR: We inserted labels referring to all subfigures.

RC: L3: I suggest '. . . represented by (b) perpendicular to and (c) along fracture'.
AR: We inserted labels referring to all subfigures.

RC: P13 Fig. 6 Why deformation does not start from zero (cf. Fig. 5).
AR: The observed crackmeter data represents the extension of the crackmeter itself. Dealing with fracture kinematics, we changed the initial deformation and set it to zero at beginning of measurements (Figure 6-8).

RC: P13 Fig. 6 caption Replace 'parallel' by 'along'?
AR: Done.

RC: P13 L4-5 Note that. . .: I cannot understand this sentence.
AR: This section was removed.

RC: P14 L3 Insert comma after 'Assuming'.
AR: Done.

RC: P14 L13-14 Hinting on. . .: Also not understandable. Lacking a verb.
AR: This paragraph has been clarified by rephrasing to: " ... TDD are not computed if the temperature time series contain a gap during summer. A weak correspondence is apparent (see Fig. 14 in appendix A) for locations with aspects to the north and east. This hints on a substantial influence of rock temperature and therefore incoming conductive energy fluxes. Interestingly, …". See 13, lines 13-15.

RC: P14 L17 Replace 'Section 5' by 'Figure 5'?
AR: We wanted to refer to the first paragraph of "Results and interpretation", which describes the evolution to the rock fall event at location *mh02* in a few sentences. We rephrased the text in the brackets to: "…  The local break-off at location *mh02* occurred in summer 2015 (described in first paragraph of Section 4, page 11).". See page 14, lines 1-2.

RC: P15 Fig. 7 Why deformation does not start from zero?
AR: This issue is addressed. See author response AR to RC: P13 Fig. 6.

RC: P18 L8-9 Which data do show 'this summer offset seems to correlate slightly with an increasing total amount of energy available (TDD)'?
AR: We added an additional figure to the appendix presenting the summer shift of kinematics perpendicular to fracture against yearly thawing degree days with a black line indicating the regression function. See Figure 14, page 23.

RC: P18 L32 What is 'the hypothesis of Hasler et al. (2012)'?
AR: Hasler et al. (2012) hypothesized a thermo-mechanically and cryogenic forcing of fracture kinematics. We addressed this issue by rephrasing the sentence to: "… Such thermo-mechanically

and cryogenic forcing of fracture kinematics has been hypothesized by Hasler et al. (2012), but their data was not fully conclusive on this point due to the short duration of the data set (1–2 years)" See 19, lines 22-24.

RC: P19 L7-8 'the water can easily drain through the strongly fractured rock and the water availability is limited': Does this situation fit any places in the rockwall? Aren't there any locations topographically favorable for water storage?
AR: The investigated field site is very steep and strongly fractured. We tried to measure water pressure in fractures without success. But we agree there might be topographically favorable spots for water storage, even close to the ridge and limited water supply. To clarify it, we rephrased this paragraph to: " If the water and/or heat supply is high enough, the water column can rise and enhance hydro pressure. But high water columns are rather unlikely at the Matterhorn field site, because it is located on the ridge with steep, laterally open fractures. " See page 19, lines 34ff.

---

## Author Comment (AC2) · 30 Sep 2016

Dear Mr. Isaksen and anonymous Referees,

We would like to thank for the detailed comments and constructive suggestions, which helped us to improve the manuscript. We hope that we have adequately addressed and answered all reviewer comments and changed the manuscript accordingly.

The referees state the topic is interesting and based on a very interesting 7 year time series of fracture displacements recorded at several locations at the Matterhorn. They highlighted several concerns which mainly concerned the clarity of the methodology, the focus and main result of the study and the introductory background information.

We addressed all these issues (and all the specific ones) raised by the referees and

briefly outline here the more substantial changes/ revisions below.

- Regarding the misinterpretation of the focus: The focus of this study is not to predicting rock slope instabilities. To address this, we clarified the focus, purpose and novelty of this study in the abstract and introduction. Regarding the weakness of the initial conceptual model: We agree, the initial conceptual model was not consistent and contained some weaknesses. To be more precise, we now use the term "fracture kinematics" instead of "fracture dynamics". We rewrote and shortened the conceptual model to an overview of the processes and related environmental controls and clarified the aim and research questions of the study in a separate section.

- Regarding difficulty in understanding methodology: We simplified and clarified the methods. We revised and clarified the whole method section. In particular, the LRM+ model was removed. Although it reproduced quite well fracture kinematics, it was not crucial for the main focus and analysis of this manuscript and could confuse readers. We also changed the term "summer offset" to "summer shift" with the abbreviation "SHT". We further extended and improved the regression analysis to investigate the relation between fracture kinematics and temperature.

- Regarding the criticism of referee 2 that a qualitative analysis of raw data would have brought the same observations/conclusions, but the proposed model does not bring significant contributions or advantages: We disagree on this point. This work provides a new quantitative analysis based on a significantly longer time series (7 years vs. 2 years). The scientific advance of this contribution is to distinguish phases as well as the timing of irreversible displacements. Timing of irreversible kinematics is crucial to link the acting mechanisms to environmental forcing. Furthermore, the developed irreversible index provides useful indication on rock wall stability.

In the revised manuscript we addressed all the reviewers' comments and added in the general response one by one explanations and comments to the specific points of the referees. We also added additional figures and changed the figures in the manuscript according to the comments.

With kind regards

Samuel Weber
On behalf of all authors

Please also note the supplement to this comment:
http://www.the-cryosphere-discuss.net/tc-2016-136/tc-2016-136-AC2-supplement.pdf
* * *
[Figure]

**Supplement:**

**Reply to comments made by Anonymous Referee #2 (doi:10.5194/tc-2016-136-RC2).**
We thank Anonymous Referee #2 for its review and suggestions for improvement. Referee comments indicated as "RC:", author reply as "AR:". Only sections requiring a reply are reproduced.

GENERAL COMMENTS
RC: The authors propose an empirical/statistical model aimed at separating reversible components of fracture deformation, due to thermo-elastic strains in alpine high-elevation permafrost environments, and the irreversible (plastic) component due to other processes. The topic is interesting and of interest for The Cryosphere. The work is based on a very interesting 7-year time series of fracture displacements recorded at several locations at the Matterhorn (Switzerland) by a monitoring network set up by Hasler et al (2012). Nevertheless, my review pointed out a number of serious scientific issues, which are listed in the following general comments and in the following "Detailed comments" section. I suggest that these points must be carefully addressed before the manuscript can be published in a high-level journal as The Cryosphere.

RC: GENERAL COMMENT 1. The abstract is quite too long and should be more focused and to-the-point;
AR: We shortened the abstract and focused more on the main results of our analysis. See page 1.

RC: GENERAL COMMENT 2. English could be generally improved using shorter and more focused sentences;
AR: We addressed this comment. The re-submitted manuscript was revised by a native speaker.

RC: GENERAL COMMENT 3. The "mechanical conceptual model" (Section 2) is characterized by some weaknesses and is not used later in the paper, which then focus on an empirical/statistical model. The authors seem to want to add a "rock mechanics taste" to the work, but tend to mix some different concepts and quantities and use terms as "fracture dynamics" which sound ambiguous to people from the geological and engineering rock mechanics communities (see detailed comments);
AR: We agree, the initial conceptual model was not consistent, contained some weaknesses and lacked clear link to the main work undertaken in this study. We replaced the conceptual model by a schematic visualization and a description of kinematics in steep fractured bedrock permafrost and the related main acting mechanisms influenced by varying environmental forcing. This part leads now more clearly to the research questions and includes the assumption for the developed linear regression model. To be more precise, we now use the term "fracture kinematics" and "fracture displacements" instead of "fracture dynamics".

RC: GENERAL COMMENT 4. The empirical/statistical model, making the core of the work, is biased by strong assumptions leading to somehow obvious results and poor predictive capability (see different detailed comments below). Actually, it is difficult for me to see either the scientific advance or the practical contribution of this work. In fact, the statistical model aims at discriminating thermo-mechanical elastic displacements, which are indeed small and of the same order of magnitude of possible precursors of rock slope instability (the latter can also follow very different patterns). This seems to suggest that the reliability of the method is low for small irreversible displacements and useless when irreversible displacements become larger. Finally, irreversible displacements are not investigated themselves thus the method cannot be used to predict rock slope failures (as promised in the abstract).
AR: We agree that the original manuscript was not clear enough on the aim and main focus. This point helped to improve the manuscript. The main results stay the same, which indicates that the previous assumptions were not invalid. We agree, the applied model is based on assumption and has limitations, but the main target is to separate reversible thermo-mechanically induced (elastic) displacements from the residual irreversible (plastic) displacements and not to predict. This model is rather a tool for fracture kinematics analysis than for prediction of rock slope failure. The focus and aim of this manuscripts are now clarified and assumptions and limitations are discussed in more detail in the revised manuscript and is investigated by a separate correlation analysis. The scientific contribution of this manuscript is to distinguish phases as well as the timing in

relation to potentially acting processes. Timing of irreversible kinematics in relation to environmental forcing is crucial for investigating and identifying the acting mechanisms and to assess rock slope stability. The results clearly show, that thermo-mechanically induced strain dominates in winter. Further, the irreversible displacements are investigated in relation of environmental forcing using the available data. This allows some inferences on potential causing mechanisms. But as referee 2 rightly points out, we can not investigate the actual causing process in detail (this point has been clarified in the manuscript).

RC: GENERAL COMMENT 5. The most interesting contribution seen here is monitoring, providing a continuous, 7-years long time series of displacements. Nevertheless, this contribution originates from the previous work by Hasler et al 2012.
AR: The data in this manuscript is based on the initial experimental and installation setup by Hasler et al. (2012). But the analysis of Hasler et al. (2012) was based on a very short time series (5 locations under 2 years and 5 locations under one year). Due to the limited duration of the data set, Hasler et al. (2012) provided only a qualitative analysis. Here, we present a much extended data set of 7 consecutive years of most sensors. Further, with this data set we undertake a much more detailed and quantitative analysis. All data used in this paper is openly available.

RC: Page 1, line 4: (and elsewhere in the manuscript): "fracture dynamics" is a confusing term to members of the rock mechanics communities (both geoscience and engineering): in fact the term "dynamics" usually refer to fracture mechanics (micro- or meso-modes of failure and related mechanical models and parameters; see e.g. Paterson & Wong) under dynamic loading conditions. Instead, the author simply refer to the temporal pattern of movements along or perpendicular to fractures. Why not use a simple term as "fracture kinematics"?
AR: We appreciate this advice. We replaced "fracture dynamics" by "fracture kinematics" or "fracture displacements" everywhere in the manuscript.

RC: Page 1, line 9: "gravity-driven slope failure": Rock slope failure? Landslide?
AR: "gravity-driven slope failure" has been removed by shortening the abstract.

RC: Page 1, lines 12-13: "enables a local assessment of rock wall stability": actually, the presented work just aims at depurate a time series of displacement along fractures from the elastic thermal component. No analysis of the spatial-temporal patterns, mechanisms and triggers of irreversible displacements is proposed, thus I do not understand how rock wall stability is dealt with here.
AR: We agree with the referee that our investigations are not focused on stability. However, the analysis includes measurements with a high temporal resolution at multiple locations with different characteristics as exposition or slope. This gives an idea of the spatial variability, but no common pattern could be detected. As irreversible kinematics can lead to instabilities, the temporal evolution of the irreversibility provides a first indication for stability assessments. We adjusted the text accordingly.

RC: Page 2, line 4: "frozen rock masses": the authors focus on rock masses with ice-filled discontinuities and exclude ice-free frozen rocks, where a thermal elastic strains indeed occur. This is ok, but I suggest that this should be declared clearly as an assumption at the beginning of the analysis, also suggesting the expected differences in the behaviors of ice-filled and ice-free rock masses with respect to slope instability. This would be very useful to non-permafrost-experts involved in the analysis of slope instability at high altitudes.
AR: In our interpretation, the adjective "frozen" refers to the aggregate state of potentially available water in a rock mass. In permafrost regions, three layers are expected. In the top layer (active layer), ice can occur seasonally if water is available. At the permafrost table (boundary between active layer and permafrost body), the percolating water freezes and stays perennially. The ice content in the permafrost body mainly depends on the water availability during permafrost aggradation. We fully agree that there are differences in the behavior of ice-filled and ice-free rock masses with respect to slope instability. But it is difficult to quantify the occurrence of ice in fractures, as the visible part of the fracture lays in the active layer and is ice-free in summer. Visual observations during field visits in winter support the seasonal availability of ice in some fractures.

RC: Page 2, line 21: "Intact high prosity rocks": and what about low porosity rocks, which form most of the Alps?
AR: We fully agree on this point, also the Matterhorn consists of low porosity rock. Unfortunately, there are limited studies investigating low porosity rocks. The same mechanism is also expected to act in rock masses with flaws in rock. We addressed this point by adding the following sentence to the manuscript: "Based on numerical simulations, ice segregation can even occur in low porosity rocks in an estimated temperature range from −4 to −15° C (Walder and Hallet, 1985)." See page 4, lines 25-26.

RC: Page 3, line 25 "thermo-elastic induced strains": the conceptual model of the authors is based on a balance of driving and resisting forces. Strains are not forces, but are related to forces by a specified rheology and geometry (i.e. Stress distribution). Balancing the contribution of strains is formally incorrect, although this has no consequence on the analysis because the mechanical model is actually not used in the following (but it is another weakness of this work; see General Comments)
AR: We agree that there was a confusing use of language/terminology in this section. We replaced the conceptual model by a schematic visualization and a description of kinematics in steep fractured bedrock permafrost and the related main acting mechanisms influenced by varying environmental forcing. This new approach built the basis for the linear regression model and the hypothesis. Based on the 7 year time series, we analyzed and discussed the influence of environmental forcing on the acting mechanisms.

RC: Page 3, line 27: "creep and fracture of ice": here the authors include among resisting forces some processes and quantities that are not forces. Creep is a time dependent deformation of materials, including a large variety of physical processes at micro to macro scales. Fracture is brittle failure of solids. I understand that ice deformation and failure reduces stresses through plastic work, but again it is formally not correct to include these processes as forces.
AR: We agree, a detailed answer is given in the previous point and the text has been revised accordingly.

RC: Page 3, line 27: "fracture infill": strength of fracture infill?
AR: Fracture infill is interpreted as a mechanism that blocks the fracture and prevents a closing of the fracture, unless there are other mechanisms which reduce the amount of infill.

RC: Page 3, line 31: "reversible and irreversible": elastic and plastic?
AR: Reversible kinematics refers to thermally-induced strain, while irreversible describes the residual kinematics. Thus, the reversible part is elastic strain, but the irreversible part can also include creep and rupture beside plastic strain. We addressed this comment by modifying the manuscript: "... The observed fracture kinematics usually consists of a reversible (elastic) and irreversible (plastic, creep and rupture) component. ..." See page 3, lines 3-4.

RC: Page 3, line 33 and Page 4, line 1: I am not convinced about the physical consistency of the "temperature-fracture deformation relationships". It is well known from a huge laboratory rock mechanics literature that the rheology (stress-strain relationships, brittle vs ductile behavior) of rocks depends on temperature. Thus, it would not be possible in principle to define unique temperature-strain relationships, especially when dealing with creep, which is non-linear and time dependent even at constant temperature. I understand that authors just refer to individual existing fracture deformations and guess that they assume linear elastic-perfectly plastic rheology in the considered temperature range. Nevertheless, the authors *should clearly state and support their assumptions and related limitations*: are they sure that stress-strain-temperature relationships for ice filled fractures (and even more for fractured rock masses!) are as simple as they state? Are they able to provide experimental data or literature to support that?
AR: We think there is a misunderstanding in scale and temperature here. The laboratory experiment of Wolters (1969) showed a linear temperature-strain relation for the temperature range from -20 to +80° C, which covers the temperature range measured at Matterhorn. Several studies in permafrost bedrock with different measurement setups (e.g. Wegmann and Gudmundsson, 1999; Matsuoka, 2001; Matsuoka and Murton, 2008; Nordvik et al., 2010) reported a simple

correlation between fracture kinematics and (rock-) temperature at different time scales from diurnal to annual. The field site Matterhorn consists of fractures with and without ice, but the stress induced by ice pressure might be limited due to the high degree of fracturing. For our model describing the reversible fracture kinematics, we assumed a linear relationship between thermo-elastic strains in rock and temperature (we modified and clarified this point in the manuscript). It is clear that reversible kinematics can not be split up in different processes, but high coefficients of determination resulting from the regression analysis indicate that it works.

RC: Page 6, line 23: "these statements are validated": in the following, the authors switch from a conceptual mechanical model to a simplified statistical one to discriminate reversible and irreversible movements along monitored fractures. Nevertheless, the postulated origins of irreversible movements, i.e. "Cryogenic" in winter and "hydro" in summer, although reasonable, are not validated by data and analysis. No information is provided about the state of ice filling in fractures, and there is no correlation between hydrological parameters (e.g. Rainfall) and irreversible movements.
AR: We agree that these statements are not explicitly validated due to limited data describing environmental conditions and no reliable data providing information about the state of ice infill in fractures is available. The paper was refocused and the hypotheses were removed, as they mainly supposed the same as the research questions.

RC: Page 6, line 29: "heterogeneous": in which sense?
AR: We addressed this point by rephrasing this sentence: "This field site consists of spatially heterogeneous steep fractured bedrock with partially debris covered ledges." See page 6, line 19.

RC: Page 6, line 30: rainfall, cold winter temperature, exposure etc.: please provide quantitative values/ranges typical of the studied environments.
AR: Unfortunately, we have limited weather data for this field site and no representative weather station of the Swiss Meteo Station Network, which is close to the field site and in a similar elevation. But we inserted the MAAT and maximum wind speed locally measured in the years 2011-2012 (see page 6, lines 19ff). We added three pictures distributed over a year (taken in the morn on 01 Jan 2015, 03 Apr 2015, 01 Jul 2015 and 01 Oct 2015) to illustrate the variability of snow deposition (see Figure 4).

RC: Page 7, line 5: could the authors explain why they measured temperature down to 85cm and not deeper? This also applies elsewhere in the manuscript. Which are the other measuring depths, and why temperature profiles are not used / presented?
AR: The depth of rock temperature measurements (0.1, 0.35, 0.6 and 0.85 m) are given by the installation of Hasler et al. 2012. The extended Table 1 on page 9 gives an overview of the available temperature in rock and fracture at different depths. A selection of the rock temperature time series are shown in Figure 6 (at the end of this reply a similar figure with temperature gradients calculated by ($T_{0.85\,m} - T_{0.1\,m}$)/0.75 m is shown). For the new analysis, temperature measurements in fractures at different depth are included. Applying a best fit analysis using all available rock and fracture temperatures, we determined the most representative temperature measurement (which are in most cases at 0.85 m depth) for modeling the reversible thermo-mechanically induced fracture kinematics. The optimized trainings windows are shown in Table 2 on page 13.

RC: Page 7, lines 5-6: "high resolution images": what are these used for, also considering that pixel resolution is of the same order of magnitude of the fracture displacements recorded in seven years?
AR: These images are mainly used for inspection of the instrumentation, but also provide information about the snow deposition. Currently, we do not derive displacements. This would be the scope of an other project.

Page 7, line 13: "aggregated": cumulated or averaged?
AR: The data was aggregated by averaging.

RC: Page 8, line 4: "Staub et al": manuscripts in review are not citable.
AR: This publication is accepted now and published as early view article.

RC: Page 8, line 9: "Used temperature ....at 85cm depth": why?
AR: This point was addressed in detail in the author response to the referee comment RC Page 7, line 5.

RC: Page 9, lines 3-10: the statistical linear model for the reversible deformations is poorly explained and supported: does it apply at the same way to shear and normal fracture displacements? How is the data population related to reversible movements separated from the irreversible movements occurring in winter for fitting purposes? Which are the best-fit statistical parameters of the model and related measures of statistical performance? These are not reported and the reader is forced to believe that the model is robust. This is a major scientific weakness of the work and the authors should work more on this.
AR: We addressed this point and explained the linear regression model in more detail. We added an additional correlation analysis for defining the trainings phase and a table with the statistical performance (Table 2, page 13). In principle, LRM can be applied the same way to shear and normal fracture kinematics, but is much more sensitive to the geometric mesoscale arrangement of the fracture. Assuming for instance the rock masses aside the fracture have the same size and thermal condition, the thermo-mechanically induced strain is also the same and no kinematics along fracture is measured. For one location (*mh08*), we added in the supplements a figure illustrating the modeled reversible, thermo-mechanically induced kinematics (Figure 13, page 22).

RC: Page 9, line 14: 28 days window length: one month is a long smoothing period, could the authors explain why they used such a long time interval? In general, one could expect that excessive smoothing may "kill" some important signals on shorter timescales.
AR: We agree that smoothing over 28 days may attenuate variations on short timescales. We adapted the irreversibility index, run the index function (Equations 3 + 4 on page 10) with a sliding windows of 21 days and do not explicitly smooth the data any further. Anyway, the irreversibility index aims at detecting periods, when the irreversible fracture kinematics dominates. On the one hand, it helps to interpret potential forcing and on the other hand, it should enable to assess the stability and not to predict rock slope instabilities.

RC: Page 10, line 11: "due to creeping": this part is obscure and, again, I cannot understand how the authors are able to separate the population of reversible vs. irreversible winter deformations.
AR: We rephrased this sentence and do not refer to a process anymore. The referee is right, we can not separate the population of reversible vs. irreversible kinematics during the training phase. We assumed that the irreversible kinematics is negligible during the trainings phase, which is confirmed by the coefficient of determination given by the regression analysis (see Table 2, page 13).

Page 10, line 21: "in winter. . ...we assume that deformation by the thermos-mechanical induced strain dominated": this indeed remains a strong assumption, possibly significantly biasing the model. The authors should try to support this better.
AR: The LRM+ model was removed. See comment above.

RC: Page 11, Figure 4 (and related text): the piecewise linear regression model sounds over-simplified and biased by different strong assumptions including the following: 1) winter deformation is always reversible (or, at least, the same reversible deformation fitted in an early "training period" occur every winter – this may be not true as the rock mass accumulates damage); 2) the beginning of the "creeping" phases can be predefined and is the same every year; 3) displacement time series in the creeping phase are linear. I suggest that these assumptions pose too many constraints on the model and hampers its application to prediction/forecasting, except in very simple cases.
AR: This figure was removed according to the explanation in RC: P10, lines 21. Instead, we analyzed the whole time series, focusing on the irreversible fracture kinematics after removing the reversible part from the raw data.

RC: Page 11, line 23: ". . .a field site can not be described by a single measurement location. . ..": this seems quite obvious, and things are even worse when dealing with rock masses instead of individual fractures.

AR: We think this statement is still valuable and very well supported by data. Individual fractures seem to respond quite differently. Multiple spatially distributed locations with different characteristics as exposition or slope, including fractured rock masses, give an idea of spatial variability. Single measurement points enable to investigate the kinematics at small scale, while an array of measurement points can help to assess the stability of the instrumented area.

RC: Page 12, lines 9-10: "indicated by a black line in Figure 6": this is unclear or incorrect. The black lines seem linear regression functions, not their coefficients (which are never reported in the paper; instead, the authors should provide tables of best-fitting function parameters and regression quality statistics or indices to demonstrate the performance of their statistical model). Moreover, it is difficult to understand why the black lines are plotted at these positions (why don't they intersect the x-axis in zero? What is actually fited?)

AR: We appreciate this note. We clarified this in the caption of Figure 7: "Black lines indicate the linear regression function determined by the regression analysis (see Table 2)." Table 2 provides the regression parameters (selected temperature, trainings phase, parameters intercept and slope of regression function, correlation coefficient and coefficient of determination).

RC: Page 13, lines 4-5: "note that. . ...deformation": incomplete statement.
AR: The LRM+ model was removed (see previous comments).

RC: Page 13, lines 5-6: "reduced data input": the authors' approach is to fit very limited time windows and then extrapolate the results. But in this way, they are not able to obtain a model fitting the entire dataset, which is particularly important to empirically fit time-dependent movements (creep). Also, in this way the potential of the beautiful 7-year presented monitoring series is not exploited.

AR: This section was removed and the full 7-year monitoring series without reduction is now discussed/explored in more detail. However, we end up with similar results showing that fracture kinematics at most locations consists of reversible thermo-mechanically induced strain, creep phase during thawing period and fracture opening in autumn when temperatures drop below 0° C.

RC: Page 14, line 1: "this likely indicates thawing related processes": this is obviously reasonable, but but still unsupported by specific analyses. "assuming that water is available. . ...deformation": same comment.

AR: We don t really understand this comment: We specifically build an index to analyze our data, and could eventually link its variations to environmental conditions. Moreover, we specifically mention this as a possible interpretation.

RC: Page 17, lines 4-5: "one single. . .. . .fracture deformation": the result is reasonable in some specific conditions (individual fracture displacements vs. rock mass, low strain, low damage, simple failure kinematics causing block movements), but is biased by the strong assumptions on which the model is based (what is reversible or irreversible deformation?)

AR: See comments above.

RC: Page 18, section 6.2: a qualitative analysis of raw data would have brought the same observations / conclusions, suggesting that the data (following the work of Hasler et al 2012) are very interesting, but the proposed model does not bring significant contributions or advantages (especially in a predictive perspective)

AR: We disagree on this point. With a qualitative analysis, it is very difficult to assess the relative contribution of reversible versus irreversible displacement and in particular the timing/evolution of irreversible displacement. This timing is however crucial in relation to the environmental forcing (melt, freezing, precipitation, …) and hence relating it to potential responsible processes.
This work provides a new quantitative analysis based on a significantly longer time series (7 years vs. 2 years). Furthermore, the developed irreversible index may be a useful measure for evaluating on rock wall stability.

---

## Referee Report (RR1)

Dear Editor,

I have reviewed the paper *Quantifying irreversible movement in steep fracture bedrock permafrost at Matterhorn (CH)* with great interest. The authors propose a statistical model, with which reversible and irreversible fracture dislocations can be decomposed. It allows inferring possible driving mechanisms from the timing of irreversible movements and may perform well as predictive modelling tool. The paper is well written and in a mature state so that I believe it can be published after some revisions and clarifications. Spell check and language check would be very helpful (I suggested some corrections, but may not have captured all errors not being a native speaker myself).

All my major and minor comments are described in the following:

Page 1 Line 13; Comma after 'Here, ....'

Line 15: 'variable rates'

Line 16: Space after '...year.'

Line 19: remove 'such'. This statement (also occurring elsewhere several times) needs to be reconsidered. What do you mean with 'water'? Water pressure? I think it is far-fetched to say that thawing or the presence of water lowers cohesion and/or friction? There might be alternative mechanisms: increased water pressure would lower the effective stress along fracture but leave the strength (i.e. cohesion and friction) untouched. However, I doubt that significant water pressure can build up in such a heavily fracture and steep, ridge-shaped topography. I would agree that thawing of ice in fractures may have an effect on strength. But how? Reducing cohesion? tensile strength? Friction? All of them? What if ice melts in a fracture that has previously been ice-filled so that the blocks were separated? If the ice melts the blocks would get into contact again and hence friction would actually be higher than with presence of ice. I suggest refining/rewording the statement to describe a mechanism that is better funded.

Line 22: '... deformation cannot be explained by a single process even at close-by locations' (check word order)

Page 2, Line 8: 'Assuming that warming ...'

Line 13: Improved monitoring strategies and hazard assessment for frozen ...'

Line 22: remove 'hereby developed'. Is it known what components change the most to increase 'shear resistance'? Cohesion or friction?

Line 28 and elsewhere: I find the term deformation for discontinuities or fractures confusing or problematic. I associate 'deformation' in rock mechanical contexts with a continuum, so a deforming fracture would be one that changes for instance shape from being planar to being curved. You are referring to movement of one side of the fracture with respect to the other one, while the fracture itself remains undeformed. I suggest using to use the term 'dislocation' for fractures (i.e. infinite deformation along a nominally flat fracture with very small aperture), and leave the term deformation for intact rock.

Line 32: remove widespread or replace by widely. Here is another instance of the term 'fracture deformation'.

Page 3, Line 1: 'unbalance'

Line 23: 'sketched' not sketched out.

Line 31: 'the observed motion'

Page 4, Line 12 - 14: The sentence is somewhat trivial as nobody expects that this equation can readily be applied. I suggest omitting.

Generally Section 2 could be shortened and written in a slightly more concise manner.

Line 22: 'stresses' not 'pressures'. Not sure that is necessarily has to lead to a 'stress reduction'. I would replace this by 'deformation/dislocation'.

Line 28: 'is' not 'get'. The sentence is not generally true. In first order, fracturing of cohesive rock bridges only stress intensity. How does the temperature dependence come in? Through presence of ice/water? The mechanism has to be explained in greater detail.

Page 6: Line 10 'It depends, among other factors,...'

Line 8: 'water' not 'hydro' (also in process D4 in Figure 1), hydrostatic pressure ('hydropressure' is not a common term).

Line 15: ' ... change the resisting forces defined by cohesion and friction ... '.

Line 16: 'e.g. from dry to wet' (Generally, it would be good check the manuscript for colloquial expressions).

Page 7: Line 8. Move sentence 'Figure 3 gives ... ' before the sentence on the measurements locations 'Fracture deformation perpendicular' ....

Also: it is not clear how dislocation parallel to a fracture is measured. I assume via extensometers spanned across fracture is an oblique manner. If that is the case, then these sensors would also measure a perpendicular component, and the parallel component has to be computed using the sensor perpendicular to the fracture. A sketch and explanation would help.

Page 8, Section 4.1: How large are the gaps? Do they occur often?

Page 9: Line 4/5: Although a smoothed temperature may resemble temperature time series a greater depth, there are phase shifts of temperature cycles towards depth. I would omit that part of the sentence.

Line 5: 'are' not 'get'.

Line 6: training window: this needs to be explained better what you mean with it.

Here also is my greatest criticism. I'm not sure if the concept of using a training window/learning period is applied in a sensible way. If the goal of the statistical model is to learn something about processes or timing of the dominant process (which I think it is the case here) it would be sufficient to calibrate the model with the entire dataset. If the goal is to demonstrate that the statistical model works as a predictive tool, it should be applied differently: to make predictions one has to use all data recorded up to a certain time, i.e. the model is calibrated against data from the start of the time series to the most recent data, and the training window is growing with time. You could for instance calibrate the model using the first 3 month, 6 month, 1 year, 2 years, etc. to show that it becomes better and better constraint or robust with time. However, choosing a training window in the middle and stating that periods in winter work better is a very arbitrary. I understand that this was done to illustrate the

robustness of the model, but it does not tell anything about its predictive capability nor is it the best calibrated model (which would be one using the entire dataset). I suggest reconsidering this calibration strategy.

Line 14: ' the difference between y and y is smoothed with a'

Section 4.4 I'm not sure if these variables do give much insight into the processes. Also the observations in Figure 8 are not very conclusive in terms of correlation between TDD and OFST. The article does not benefit much from it. However, it is up to the author if they leave it in or not.

Page 10, Line 27: 'There are two options for the end...', 'when the rock temperature crosses'

The choice of the end and start of the reversible period sound somewhat arbitrary. It relies on the preassumption that the irreversible period only occurs in summer or when temperatures are above  $-1^{\circ}$ . Later this assumption is sold as a result / conclusion drawn from the data. A different strategy would be to let data tell, when to set the start / end of the irreversible period. The irreversibility index offers itself to guide the onset and end of the period.

Replace 'get' or 'got' by proper passive tense.

Page 11, Line 22-25: 'the instrumented rock', 'the observed fracture deformation'

Line 22: Order of Figures: here Figure 10 follows Figure 5.

Page 12, Line 14: thermo-elastic would be a more appropriate term to talk about a reversible process. (also elsewhere)

Page 13, Line 5: check sentence, there is something wrong here.

Page 14, Line 7: 'thermo-elastic' instead of 'therm-mechanical'

Line 12: 'distinct'

Page 17, Line 7: 'melt onset'

Line 5: what is the reason that mh02 does not show any temperature-dependent reversible movement? I think it is remarkable that a fracture does not react on temperature! Do the authors know the reason? I think an explanation would be warranted.

Page 18, Line 6: 'rates'

Line 15: 'cannot' not 'can not' also elsewhere.

Line 19: Not only at mh02 and mh21 does OFST and TDD not correlate. How about mh08, or the last point at mh03?. As mentioned earlier I am not too convinced about the value of these observations. Maybe if more explanation/analysis is offered it could be a good contribution to process understanding. However, you do not really elaborate much on the different behaviours.

Line 25: How many days before the rock fall did this increase occur? I think even without the irreversibility index the change in behaviour was readily visible from the fracture opening data. Do you know the volume of the break-off?

Line 32 'was not observed to close'

Page 19, Line 20: During additional phases, ....

Line 22: 'suggesting a decrease of cohesion and friction'  $\rightarrow$  as mentioned earlier this is a too farfetched conclusion and is not directly supported by your data. It may also be that during summer, stress redistribute such that strength (i.e. friction and cohesion) is overcome and slip initiates, while friction and cohesion themselves do not change.

If you compare with

Collins, B. D. & Stock, G. M. Nature Geosci. http://dx.doi. org/10.1038/ngeo2686 (2016).

or

Gunzburger, Y., Merrien-Soukatchoff, V. & Guglielmi, Y. Int. J. Rock Mech. Min. Sci. 42, 331–349 (2005).

irreversible fracture opening or slip does not have to be related to a change in strength (or not even to ice, although it may well be the case). To me it is not entirely clear by what mechanism irreversible movements in your case are induced: are the tensile fractures 'glued' with ice in winter (in this case it would be a change in tensile strength) or is it ice along sliding planes? Can something be deduced from your data and structural observations (block shapes, fracture orientations?)

Line 24-25: I suggest omitting this sentence. It does not conclude from your observations.

I hope that these comments help for improving the presentation of this very interesting work. If you have questions you would like to discuss in person, you are most welcome to contact me in person by email or phone.

Best regards,

Valentin Gischig ETH Zürich Sonneggstrasse 5, NO F27 CH-8092 Zürich Phone: +41 44 632 36 48 E-Mail: gischig@erdw.ethz.ch

---

## Author Response (AR2)

**General comments and response by authors**

Dear Mr. Isaksen, Mr. Gischig and anonymous Referee,

We would like to thank for the detailed comments and constructive suggestions, which helped us to improve the manuscript. We hope that we have adequately addressed and answered all referee's comments and changed the manuscript accordingly.

In the revised manuscript we addressed all the referees' comments and added in the general response explanations and comments to the specific points of the referees. The comments made by Referee #4 related to training window/learning period refers to the Initial Submission and was strongly improved after the 1st Revised Submission. We tried to answer this comment satisfactorily. We addressed and clarified the remaining comments that are still relevant in the second revision. We also changed the figures in the manuscript according to the comments.

With kind regards

Samuel Weber

On behalf of all authors

**Reply to comments made by Anonymous Referee #3**
We thank Anonymous Referee #3 for its review and suggestions for improvement. Referee comments indicated as "RC:", author reply as "AR:". Only sections requiring a reply are reproduced.

RC: In the introduction, it may be appropriate to add a short paragraph/sentence about microseismic monitoring and fracture development, as this topic is also cited by the authors themselves in the last paragraph of the conclusions. As far as references are concerned, apart from the work by Murton and Matsuoka, I would consider: 1) Occhiena C, Coviello V, Arattano M, Chiarle M, Morra di Cella U, Pirulli M, Pogliotti P, Scavia C (2012) Analysis of microseismic signals and temperature recordings for rock slope stability investigations in high mountains areas. Natural Hazards and Earth System Sciences. 12: 2283-2298; 2) Arosio D, Longoni L, Mazza F, Papini M, Zanzi L (2013) Freeze-thaw cycle and rockfall monitoring. In: Margottini C et al (ed's) Landslide Science and Practice, Vol.2, Springler, Berlin Heidelbergh, p 385-390. The first paper describes the relationship between acoustic emission and temperatures on the Matterhorn, while the second paper presents interesting lab tests, considering the role of ice expansion.
AR: Micro-seismic monitoring is for sure a way to complement the present study. In the recently published paper (Murton et al., 2016) 1000 micro-seismic events coincident to rock fracturing in a three year freezing experiment were analyzed and clustered according to presumable fracturing types (crack coalescence, initial fracturing…). A similar setup could in future reveal insights into relevant fracturing types. We only mentioned this method in outlook of this manuscript. This will be the scope of another paper that intends to link micro-seismic activity and irreversibility index given by the analysis of crackmeter measurements. Adding a short paragraph/sentence about micro-seismic might not be pertinent in the introduction section and be confusing for the reader, as we intend to separate and quantify the irreversible displacement only with crackmeter and temperature measurements. However we rephrased the last paragraph of the conclusion (page 20, line 23).

Murton, J., Kuras, O., Krautblatter, M., Cane, T., Tschofen, D., Uhlemann, S., Schober, S., Watson, P., 2016. Monitoring rock freezing and thawing by novel geoelectrical and acoustic techniques. Journal of Geophysical Research – Earth Surface.

RC: Page 3, line 5. Typo
AR Done.

RC: I would spilt Fig. 1 in sub-figures with proper labels and I would refer to them in the following paragraphs.
AR: Done.

RC: Page 4, line 4. "This is therefore a reversible mechanism". The cause and effect relationship is not clear here. Could you please explain in more details?
AR: To clarify this point, we modified the sentence in the revised manuscript to (page 4, line 7): "This is therefore a reversible mechanism as it is driven by cycling temperature."

RC: Page 5, line 16. What about the water lubricating the fractures? Could you comment on that?
AR: It seems that the lubrication mechanisms are investigated for fault rock by earthquakes. The effect of water lubricating the fractures is ambiguously discussed in the literature and therefore not included here. However, we don't think that it is a dominant and relevant fact in such a field site. But the presence of water can reduce cohesion in clay, or possibly also rates of critically stressed fracture propagation in intact rock. We addressed this comment by rephrasing the paragraph (page 5, line 16):
"However, changing conditions in shear zones, e.g. from dry to wet, can lead to irreversible displacement, for example caused by water (melting snow or rain) percolating through preexisting fissures. Even with low hydrostatic pressure, the presence of water can reduce cohesion in fine-grained material containing clay and is expected to have a strong influence in fractures filled with fine-grained material."

RC: Page 5, line 24. I would change into "could be assumed to be". In some failures no

displacements are observed before ultimate collapse. Please consider also modifying sentence at page 6, lines 6-7.
AR: Done.

RC: Page 6, line 2. "(middle part of Fig. 1)". Not clear. What do you mean?
AR: We modified the figure labeling in the revised manuscript and refer to Figure 1a.

RC: Page 6, line 17. What do you mean with "obvious"? Please clarify.
AR: We clarified this point by rephrasing the paragraph (page 6, line 15).

RC: Page 7, caption Fig. 3. What is an active layer of the permafrost?
AR: To clarify this point, we added a definition of active layer in the caption of Fig. 1 (page 3).

RC: Page 8, line 6. Information is singular.
AR: Done

RC: Page 9, line 5. Could you add a reference for the Pearson correlation?
AR: Done.

RC: Page 10, line 15. Why 21 days? Please explain.
AR: The length of the sliding window of 21 days was defined iteratively as a trade off between high noise-level and loosing important signals due to smoothing (page 10, line 14).

RC: Page 11, line 2. "rises".
AR: Done

RC: Page 11, line 12. Please change into: "are not visible after mid 2015 as they are out of range (Fig. 6)."
AR: Done

RC: Page 11, line 13. Is 18 May 2015 early summer?
AR: To clarify this, we rephrased this sentence in the revised manuscript to (page 11, line 12): "This abrupt and large displacement is due to a small rock fall event with a volume of a few cubic meters on 18 May 2015."

RC: Page 11, line 22. "according to".
AR: Done

RC: Page 11, line 25. "exhibit".
AR: Done

RC: Page 12, lines 1-2. Please rephrase this sentence.
AR: We rephrased the beginning of the caption to (caption Fig. 6, line 12): "Thermal conditions and fracture displacements at the Matterhorn Hörnligrat field site over a course of eight years".

RC: Page 20, line 17. "superimposed on".
AR: Done

**Reply to comments made by Referee #4 Valentin Gischig.**

We thank Valentin Gischig (Referee #4) for his review and suggestions for improvement. Referee comments indicated as "RC:", author reply as "AR:". Only sections requiring a reply are reproduced.

Page 1 Line 13; Comma after 'Here, ....'
AR: Rephrased in the revised manuscript.

RC: Line 15: 'variable rates'
AR: Rephrased in the revised manuscript.

RC: Line 16: Space after '...year.'
AR: Done.

RC: Line 19: remove 'such'. This statement (also occurring elsewhere several times) needs to be reconsidered. What do you mean with 'water'? Water pressure? I think it is far-fetched to say that thawing or the presence of water lowers cohesion and/or friction? There might be alternative mechanisms: increased water pressure would lower the effective stress along fracture but leave the strength (i.e. cohesion and friction) untouched. However, I doubt that significant water pressure can build up in such a heavily fracture and steep, ridge-shaped topography. I would agree that thawing of ice in fractures may have an effect on strength. But how? Reducing cohesion? tensile strength? Friction? All of them? What if ice melts in a fracture that has previously been ice-filled so that the blocks were separated? If the ice melts the blocks would get into contact again and hence friction would actually be higher than with presence of ice. I suggest refining/rewording the statement to describe a mechanism that is better funded.
AR: We agree to this point, it is not known how the availability affects cohesion or friction. This statement in the initial submission was reconsidered and clarified for the revised manuscript. Significant water pressures can build up even in fractured rock masses above permafrost bodies as perched water above ice-sealed fractures (Pogrebiskiy and Chernyshev, 1977) but there are no detailed empirical quantitative studies on how hydrostatic pressure affects rock walls in permafrost regions. Ice in fractures influences shear resistance due to creep and fracturing of ice itself and along rock-ice interfaces (Krautblatter et al., 2013) and produce tensile strength of typically up to 2 MPa. The performance of ice is controlled by stress, temperature and water/impurity content in the ice.

RC: Line 22: '... deformation cannot be explained by a single process even at close-by locations' (check word order)
AR: Rephrased in the revised manuscript.

RC: Page 2, Line 8: 'Assuming that warming ...'
AR: Done.

RC: Line 13: Improved monitoring strategies and hazard assessment for frozen ...'
AR: Done.

RC: Line 22: remove 'hereby developed'. Is it known what components change the most to increase 'shear resistance'? Cohesion or friction?
AR: We rephrased this statement to (page 4, line 31): "While ice-filled joints can form relatively tough ice bodies at low temperatures, the shear resistance decreases with rising temperature and reaches a minimum just below the thawing point (Davies et al., 2001)." The study of Davies only considers change in temperature and normal stress and does not provide further information concerning relative change in the different component (cohesion or friction).

RC: Line 28 and elsewhere: I find the term deformation for discontinuities or fractures confusing or problematic. I associate 'deformation' in rock mechanical contexts with a continuum, so a deforming fracture would be one that changes for instance shape from being planar to being curved. You are referring to movement of one side of the fracture with respect to the other one,

while the fracture itself remains undeformed. I suggest using to use the term 'dislocation' for fractures (i.e. infinite deformation along a nominally flat fracture with very small aperture), and leave the term deformation for intact rock.

AR: Dislocation is generally used in materials science to describe a defect within a crystal structure. Replacing deformation by dislocation would be therefore confusing for most of readers. However, reviewer is right, deformation in the context of discontinuities can be problematic. We therefore clarified this point by using the term fracture displacement and defining our terminology (page 6, line 6): "The term displacement used in the following refers to the movement of one side of the fracture with respect to the other."

RC: Line 32: remove widespread or replace by widely. Here is another instance of the term 'fracture deformation'.
AR: Done.

RC: Page 3, Line 1: 'unbalance'
AR: Rephrased in the revised manuscript.

RC: Line 23: 'sketched' not sketched out.
AR: Rephrased in the revised manuscript.

RC: Line 31: 'the observed motion'
AR: Rephrased in the revised manuscript.

RC: Page 4, Line 12 – 14: The sentence is somewhat trivial as nobody expects that this equation can readily be applied. I suggest omitting. Generally Section 2 could be shortened and written in a slightly more concise manner.
AR: Rephrased in the revised manuscript.

RC: Line 22: 'stresses' not 'pressures'. Not sure that is necessarily has to lead to a 'stress reduction'. I would replace this by 'deformation/dislocation'.
AR: We rephrased this sentence to (page 5, line 23): "Deformation and fracture of ice can absorb stress along fractures and lead to dislocation...".

RC: Line 28: 'is' not 'get'. The sentence is not generally true. In first order, fracturing of cohesive rock bridges only stress intensity. How does the temperature dependence come in? Through presence of ice/water? The mechanism has to be explained in greater detail.
AR: We agree and removed this sentence as it is over simplified.

RC: Page 6: Line 10 'It depends, among other factors,...'
AR: Done.

RC: Line 8: 'water' not 'hydro' (also in process D4 in Figure 1), hydrostatic pressure ('hydropressure' is not a common term).
AR: We removed the term hydropressure and replaced it by hydrostatic pressure.

RC: Line 15: ' ...change the resisting forces defined by cohesion and friction ... '.
AR: Rephrased in the revised manuscript.

RC: Line 16: 'e.g. from dry to wet' (Generally, it would be good check the manuscript for colloquial expressions).
AR: Done.

RC: Page 7: Line 8. Move sentence 'Figure 3 gives ... ' before the sentence on the measurements locations 'Fracture deformation perpendicular' .… Also: it is not clear how dislocation parallel to a fracture is measured. I assume via extensometers spanned across fracture is an oblique manner. If that is the case, then these sensors would also measure a perpendicular component, and the parallel component has to be computed using the sensor perpendicular to the fracture. A sketch

and explanation would help.

AR: We changed the order of the sentences. In the revised manuscript, we adapted Figure 5 (page 8) and added a photo with a sketch that illustrates locations instrumented with two crackmeters.

RC: Page 8, Section 4.1: How large are the gaps? Do they occur often?

AR: We clarified this in the caption of Figure 6 (page 12)in the revised manuscript: "A gap in the rock temperature time series of location mh12 ($T_{east}$) is filled for the time period November 2012 until July 2013 and from August 2014 onwards applying quantile mapping using the best regressors approach (Staub et al., 2016) with a coefficient of determination $R^2 = 0.92$."

RC: Page 9: Line 4/5: Although a smoothed temperature may resemble temperature time series a greater depth, there are phase shifts of temperature cycles towards depth. I would omit that part of the sentence.

AR: This part was rephrased in the revised manuscript.

RC: Line 5: 'are' not 'get'.

AR: Rephrased in the revised manuscript.

RC: Line 6: training window: this needs to be explained better what you mean with it. Here also is my greatest criticism. I'm not sure if the concept of using a training window/learning period is applied in a sensible way. If the goal of the statistical model is to learn something about processes or timing of the dominant process (which I think it is the case here) it would be sufficient to calibrate the model with the entire dataset. If the goal is to demonstrate that the statistical model works as a predictive tool, it should be applied differently: to make predictions one has to use all data recorded up to a certain time, i.e. the model is calibrated against data from the start of the time series to the most recent data, and the training window is growing with time. You could for instance calibrate the model using the first 3 month, 6 month, 1 year, 2 years, etc. to show that it becomes better and better constraint or robust with time. However, choosing a training window in the middle and stating that periods in winter work better is a very arbitrary. I understand that this was done to illustrate the robustness of the model, but it does not tell anything about its predictive capability nor is it the best calibrated model (which would be one using the entire dataset). I suggest reconsidering this calibration strategy.

AR: We addressed the selection of the trainings phase in the revised manuscript. We added an additional correlation analysis for defining the trainings phase. We applied a best fit analysis using all available rock and fracture temperature data. Due to complete data availability at all instrumented locations, only the data in the time window between 1 Oct 2013 and 1 Jan 2015 is considered. We determined on this period the most representative temperature measurement for modeling the reversible thermo-mechanically induced fracture kinematics. The best trainings periods are shown in Table 2 on page 13. In our opinion it does not make sense to calibrate the model with the entire data set as the model only describe thermo-elastic strain. The correlation analysis shows that the coefficient of determination decreases if the training phase is too long. High coefficients of determination show that there are time periods dominated by thermo-elastic strains.

RC: Line 14: ' the difference between y and y is smoothed with a'

AR: This point was addressed in the revised manuscript.

RC: Section 4.4 I'm not sure if these variables do give much insight into the processes. Also the observations in Figure 8 are not very conclusive in terms of correlation between TDD and OFST. The article does not benefit much from it. However, it is up to the author if they leave it in or not.

AR: We added an additional figure to the appendix of the revised manuscript presenting the summer shift of kinematics perpendicular to fracture against yearly thawing degree days with a black line indicating the regression function (see Figure 14, page 23). We clarified this paragraph by rephrasing to: " ... TDD are not computed if the temperature time series contain a gap during summer. A weak correspondence is apparent (see Fig. 14 in appendix A) for locations with aspects to the north and east. This hints on a substantial influence of rock temperature and therefore

incoming conductive energy fluxes. Interestingly, …".  (page 13, line 13)

RC: Page 10, Line 27: 'There are two options for the end...', 'when the rock temperature crosses' The choice of the end and start of the reversible period sound somewhat arbitrary. It relies on the pre-assumption that the irreversible period only occurs in summer or when temperatures are above -1°. Later this assumption is sold as a result / conclusion drawn from the data. A different strategy would be to let data tell, when to set the start / end of the irreversible period. The irreversibility index offers itself to guide the onset and end of the period. Replace 'get' or 'got' by proper passive tense.
AR: We revised and clarified the whole method section in the revised manuscript. In particular, the LRM+ model was removed. Although it reproduced quite well fracture kinematics, it was not crucial for the main focus and analysis of this manuscript and could confuse readers.

RC: Page 11, Line 22-25: 'the instrumented rock', 'the observed fracture deformation'
AR: Done.

RC: Line 22: Order of Figures: here Figure 10 follows Figure 5.
AR: The order of the figures is in this paragraph not consecutive as the first figure refers to the figure in the attachment.

RC: Page 12, Line 14: thermo-elastic would be a more appropriate term to talk about a reversible process. (also elsewhere)
AR: We rephrased all terms "thermo-mechanically induced strain" by "thermo-elastic strain".

RC: Page 13, Line 5: check sentence, there is something wrong here.
AR: This paragraph was rephrased in the revised manuscript.

RC: Page 14, Line 7: 'thermo-elastic' instead of 'therm-mechanical'
AR: Done.

RC: Line 12: 'distinct'
AR: Done.

RC: Page 17, Line 7: 'melt onset'
AR: Done.

RC: Line 5: what is the reason that mh02 does not show any temperature-dependent reversible movement? I think it is remarkable that a fracture does not react on temperature! Do the authors know the reason? I think an explanation would be warranted.
AR: A possible explanation could lie in the individual geometric mesoscale arrangement of each fracture. Actually we do not know for sure, but we guess that the fracture is more inclined and the thermo-elastic strain of the rock masses aside the fracture is detectable at the outer boundary of the rock mass. In this case, displacement occurs, but is not measured by the installed sensor setup. As this explanation is strongly hypothetical, we extended the paragraph with (page 18, line 18):
"..., the magnitude of the reversible fracture displacement, caused by thermo-elastic strain, is influenced by the individual geometric mesoscale arrangement of each fracture."

RC: Page 18, Line 6: 'rates'
AR: We rephrased this paragraph.

RC: Line 15: 'cannot' not 'can not' also elsewhere.
AR: Done.

RC: Line 19: Not only at mh02 and mh21 does OFST and TDD not correlate. How about mh08, or the last point at mh03?. As mentioned earlier I am not too convinced about the value of these

observations. Maybe if more explanation/analysis is offered it could be a good contribution to process understanding. However, you do not really elaborate much on the different behaviours.
AR: See author response to reviewer comment "RC: Section 4.4"

RC: Line 25: How many days before the rock fall did this increase occur? I think even without the irreversibility index the change in behaviour was readily visible from the fracture opening data. Do you know the volume of the break-off?
AR: We rephrased this paragraph in the revised manuscript (page 11, line 12): "This abrupt and large displacement is due to a small rock fall event with a volume of a few cubic meters on 18 May 2015. The functionality of both crackmeters was however not affected. But the thermistors at location mh02 were damaged by falling rocks. Hence the temperature time series ends on 18 May 2015. After this rock fall event, the fracture at location mh02 continued to deform in several small steps until late summer (14 August 2015) when the instrumented rock broke off completely during a bad weather period (see Fig. 12)." Unfortunately, we do not know the exact volume of the break-off.

RC: Line 32 'was not observed to close'
AR: This paragraph was rephrased in the revised manuscript.

RC: Page 19, Line 20: During additional phases, ....
AR: Done.

RC: Line 22: 'suggesting a decrease of cohesion and friction' ⊐ as mentioned earlier this is a too far-fetched conclusion and is not directly supported by your data. It may also be that during summer, stress redistribute such that strength (i.e. friction and cohesion) is overcome and slip initiates, while friction and cohesion themselves do not change. If you compare with
Collins, B. D. & Stock, G. M. Nature Geosci. http://dx.doi. org/10.1038/ngeo2686 (2016).
or
Gunzburger, Y., Merrien-Soukatchoff, V. & Guglielmi, Y. Int. J. Rock Mech. Min. Sci. 42, 331–349 (2005).
irreversible fracture opening or slip does not have to be related to a change in strength (or not even to ice, although it may well be the case). To me it is not entirely clear by what mechanism irreversible movements in your case are induced: are the tensile fractures 'glued' with ice in winter (in this case it would be a change in tensile strength) or is it ice along sliding planes? Can something be deduced from your data and structural observations (block shapes, fracture orientations?)
AR: This comment refers to the initial submission and was rephrased for the first Revised Submission. Regarding the mechanism leading to irreversible displacement: Anyway, reviewer might be right, if , for any reason, local stress overcome strength (even constant), a slip might occur (i.e. irreveversible displacement). But based on only surface displacement and temperature measurements, it is difficult to decipher the process leading to irreversible displacement.

RC: Line 24-25: I suggest omitting this sentence. It does not conclude from your observations.
AR: This paragraph was rephrased in the revised manuscript.

[revised manuscript text omitted]